



# Determination of the Emission Rates of CO₂ Point Sources with Airborne Lidar

Sebastian Wolff[1], Gerhard Ehret[1], Christoph Kiemle[1], Axel Amediek[1], Mathieu Quatrevalet[1], Martin Wirth[1], and Andreas Fix[1]

[1]Deutsches Zentrum für Luft– und Raumfahrt (DLR), Institut für Physik der Atmosphäre, Oberpfaffenhofen, Germany

*Correspondence to*: Sebastian Wolff (sebastian.wolff@dlr.de)

**Abstract.** Anthropogenic point sources, such as coal-fired power plants, produce a major share of global $CO_2$ emissions. International climate agreements demand their independent monitoring. Due to the large number of point sources and their global spatial distribution, a mobile measurement approach with fast spatial coverage is needed. Active remote sensing measurements by airborne lidar show much promise in this respect. The integrated-path differential-absorption lidar CHARM–F is installed onboard an aircraft, in order to detect weighted vertical columns of $CO_2$ mixing ratios, below the aircraft along its flight track. During the Carbon Dioxide and Methane mission (CoMet) in spring 2018, airborne greenhouse gas measurements were performed, focusing on the major European sources of anthropogenic $CO_2$ emissions, i.e. large coal–fired power plants. The flights were designed to transect isolated exhaust plumes. From the resulting enhancement in the $CO_2$ mixings ratios, emission rates can be derived in terms of the cross–sectional flux method. On average, we find our results roughly corresponding to reported annual emission rates, but observe significant variations between individual overflights ranging up to a factor of 2. We suppose that these variations are mostly driven by turbulence. This hypothesis is supported by a high–resolution large eddy simulation that enables us to give a qualitative assessment of the influence of plume inhomogeneity on the cross–sectional flux method. Our findings suggest avoiding periods of strong turbulence, e.g. midday and afternoon. More favorable measurement conditions prevail during nighttime and morning. Since lidars are intrinsically independent of sunlight, they have a significant advantage in this regard.

## 1 Introduction

$CO_2$ causes the strongest radiative forcing among all anthropogenic greenhouse gases (e.g. Myhre et al. (2014)). It therefore plays a crucial role with respect to human-induced climate change. In 2018, $CO_2$ has reached a global annual average of 407.4 ppm at the Earth's surface, an increase of 47 % compared to the year ~1750 (Friedlingstein et al., 2019). One-third of all anthropogenic $CO_2$ emissions stem from localized point sources, in particular coal–fired power plants (Oda and Maksyutov, 2011). For Europe they even account for 45% of $CO_2$ emissions (Super et al., 2020). The Paris Climate Agreement aims to reduce anthropogenic greenhouse gas (GHG) emissions by means of *nationally determined contributions* (NDCs), which are based on national capabilities and the level of economic development (UNFCCC, 2015). Therein it is foreseen that as of 2023



a *global stocktake* will take place every 5 years. This requires independent measurements to verify each nation's emission reports of $CO_2$, but also of other greenhouse gases, such as $CH_4$. Currently, there is no independent global emission verification system available, and a complete record of all emissions globally is still far from reality. To achieve this goal, satellite missions are indispensable. Satellite missions are expected to detect $CO_2$ emissions from large power plants and cities, e.g., the future European Carbon Constellation CO2M (Kuhlmann et al., 2020; Broquet et al., 2018; Bézy et al., 2019), and other mission

ideas still in the pre-development phase (Kiemle et al., 2017; Strandgren et al., 2020). Furthermore, $CH_4$ emissions can also be detected, as is done by GHGSat–D for coal mine ventilation shafts (Varon et al., 2020), or the Sentinel-5 Precursor for the oil and natural gas producing sector (Zhang et al., 2020; Pandey et al., 2019). However, at the moment no operating satellite mission is able to reliably quantify emissions from large power plants. In the development phase for potential missions, airborne measurement campaigns serve as a test of the methods. During the operating phase they are needed for verification

of the space-borne results.

In May/June 2018, the CoMet (Carbon Dioxide and Methane mission) field campaign took place. Besides supporting activities for GHG-stocktaking, the objectives of CoMet are to investigate the fluxes of the major human-influenced GHG on local, regional, and sub–continental scales, and to determine them more precisely than previously possible. The CoMet campaign saw the deployment of a suite of the most sophisticated airborne instruments to measure atmospheric $CH_4$ and $CO_2$, as well as

a variety of ground–based instruments. In particular, also the synergetic use of active remote sensing (lidar) (Amediek et al., 2017; Wildmann et al., 2020), passive spectrometry (Luther et al., 2019; Krings et al., 2011), and in situ measurements (Fiehn et al., 2020; Gałkowski et al., 2020) supported by modelling activities (Chen et al., 2020; Nickl et al., 2020), as well as the validation of existing (e.g. Sentinel–5P, GOSAT (Greenhouse Gases Observing Satellite)) and the preparation of upcoming (e.g. MERLIN (Methane Remote Sensing Lidar Mission)) GHG satellite missions were aimed at.

Hereby, the German research aircraft HALO (High Altitude and Long Range Research Aircraft) acted as the airborne flagship of that campaign. HALO was equipped with the new airborne $CO_2$ and $CH_4$ IPDA (integrated-path differential-absorption) lidar CHARM–F ($CO_2$ and $CH_4$ Remote Monitoring–Flugzeug) built and operated by DLR as an airborne demonstrator for the upcoming MERLIN mission (Ehret et al., 2017). CHARM–F simultaneously measures the column–averaged dry–air mixing ratios of carbon dioxide ($XCO_2$) and methane ($XCH_4$) between aircraft and ground (Amediek et al., 2017). The influence

of other trace gases, in particular $H_2O$, on the mixing ratio measurements is negligible. The high laser pulse repetition rate of 50 Hz (double pulse) and the small divergence (~1.5 mrad) enable a dense sequence of footprints on the ground. The vertical column measurements are insensitive to vertical redistribution of the trace gases. The insensitivity towards optically thin clouds, aerosol layers, and varying surface albedo, and the sophisticated instrument design with e.g. active laser frequency control is a further strong asset of the IPDA lidar approach. Albedo variations basically influence the measurement precision

(statistical uncertainty), not the bias. Moreover, it is possible to apply adaptive averaging, whereby the measurement precision can strongly be improved.

During the CoMet campaign, HALO probed various local plumes of different coal-fired power plants. As a case study the paper in hand focuses on the measurement flight of 23 May 2018, where the $CO_2$ exhaust plume of the power plant



Jänschwalde, close to the Polish-German border was surveyed. The specific goal is to quantify the $CO_2$ fluxes of the power

plant. An established method for quantifying emission rates of point sources is the *cross-sectional flux method*, which is a product of mean wind speed and an integrated concentration enhancement along a cross-sectional overflight of the exhaust plume. This principle has been applied for air-/space-borne nadir-viewing remote sensing (Krings et al., 2018; Menzies et al., 2014; Varon et al., 2018), mobile ground based sun–viewing remote sensing (Luther et al., 2019), as well as airborne in situ measurements (Cambaliza et al., 2014; Conley et al., 2016; Fiehn et al., 2020; White et al., 1976). Amediek et al. (2017) have

described how this principle can be realized with CHARM–F. Using CHARM–F data from the respective overflights, we strive to accurately assess the error, and advance the general methodology.

When determining the cross-sectional flux, one of the major error sources is the local wind field. On the one hand, because the wind speed is directly included in the calculation, on the other hand, because atmospheric turbulence can broaden or constrict the spatial extent of the exhaust plume. This is a well-known problem, which contributes significantly to the measurement

error (Kuhlmann et al., 2019; Luther et al., 2019; Strandgren et al., 2020; Varon et al., 2018; Jongaramrungruang et al., 2019; Kumar et al., 2020). Consequently, the observed $CO_2$ column enhancements between subsequent plume transects may vary considerably, despite a constant emission rate. We hypothesize that these turbulence-induced variations dominate the measurement error of the emission rates, rather than the GHG column measurement uncertainty itself. To assess the impact of this atmospheric turbulence on our measurement results, we perform a large eddy simulation (LES) in order to resolve local

plume structures. By doing so, we are able to compare different ambient weather and turbulence conditions. Our aim is to separate more and less favorable conditions, to determine an adequate distance between emission source and measurement locations, and to find out how many independent plume measurements will be necessary in order to obtain an appropriate emission rate accuracy, as function of those environmental conditions.

This paper is organized as follows: Section 2 introduces the IPDA lidar method, describes the retrieval of the emission rate,

and the methodical errors. Section 3 reports on the plume measurement results. Section 4 provides the simulation setup, while the subsequent results are presented in Sect. 5. A discussion is given in Sect. 6, while a conclusion and outlook can be found in Sect. 7.

## 2 Cross-sectional flux method

### 2.1 Flux calculation

The dataset underlying this work originates from IPDA lidar CHARM–F. A more detailed description of the lidar system can be found in Amediek et al. (2017). At its core, an IPDA lidar consists of a high-power laser and a detector. Installed on an aircraft or satellite, the nadir-oriented lidar emits two laser pulses that propagate through the atmosphere until they are backscattered at a surface. The two backscattered laser pulses are detected by the lidar. The wavelength of one laser pulse corresponds to the absorption wavelength of the greenhouse gas under consideration. In the following, this laser pulse is

referred to as online. Due to molecular absorption the intensity of the online laser pulse decreases while propagating through



the atmosphere: The wavelength of the other (offline) laser pulse is slightly shifted such that almost no absorption by the greenhouse gas takes place, but the interaction with the remaining atmospheric components is unaltered.

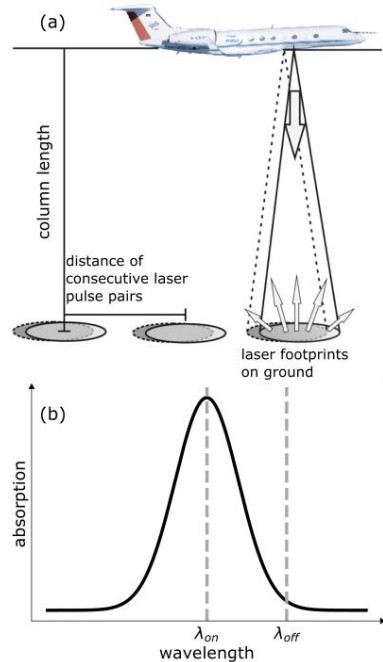

Figure 1: Figure by Amediek et al. (2017). The measurement geometry of a lidar, illustrated by the example of CHARM–F carried on the aircraft HALO. Two laser pulses are emitted towards the earth with a delay of 500 µs. The laser pulse with the wavelength $\lambda_{on}$ is on the absorption wavelength of $CO_2$ (1572.02 nm), the laser pulse with the wavelength $\lambda_{off}$ is not (1572.12 nm). By comparing the backscattered intensities, the $CO_2$ concentration in the measured cone volume can be calculated. The measured volume is usually referred to as a vertical air column, since the column length, i.e. the aircraft's altitude above ground (~ 6500 m), is very large compared to the diameter of the reflecting surfaces (~ 10 m). The distance of consecutive laser pulse pairs is 3 m.

Using a beam splitter, a small part of both the online and the offline laser pulse energy ($E_{on/off}$) is deflected onto a detector while still in the lidar system. Together with the optical power entering the lidar telescope $P_{on/off}$ the differential optical

absorption depth (DAOD) can by calculated:

$$DAOD = \frac{1}{2} \cdot ln\left(\frac{P_{off}/E_{off}}{P_{on}/E_{on}}\right) \qquad (1)$$

Note that for the DAOD a single value is obtained for the entire vertical air column. It is a metric for the greenhouse gas concentration of the measured column and is also defined by the following relationship:

$$DAOD = \frac{\Delta\sigma}{M} \int_0^{fl} c(z)\, dz \qquad (2)$$

Here, $\Delta\sigma$ is the difference between the absorption cross section of the two laser pulses given in square meter (cf. Fig. 1). It is referred to as the differential–absorption cross section and considered constant over the vertical plume extent. This is discussed





in more detail in Sect. 3. M is the molecular mass of $CO_2$ in gram and $c(z)$ is the $CO_2$ density in gram per cubic meter. The vertical integral limits are the ground ($z = 0$ m) and the respective height of the aircraft fl. Variations in flight altitude as well as topography may cause variations in the surveyed column length and thus ultimately in the measured DAOD. In this study

these variations are negligible, since on the one hand the flight altitude was deliberately kept constant, and on the other hand the topography around the power plant under consideration is sufficiently flat.

This DAOD dataset is used to determine the $CO_2$ emission rate of a point source by means of the flux calculation method introduced by Amediek et al. (2017). As schematically depicted in Fig. 2, a crossing of the exhaust plume leads to a DAOD enhancement. This is caused by additional absorption of laser radiation by the $CO_2$ molecules of the plume. The instantaneous

flux through the lidar cross-section, in the moment of the overflight, is given in kilogram per second and denoted by q:

$$q = A \cdot \frac{M}{\Delta\sigma} \cdot u \cdot sin(\varphi) \tag{3}$$

Given in meter, the parameter A corresponds to the integrated DAOD enhancement over the background DAOD in the direction of the aircraft flight track as shown in Fig. 2b. In the following it is referred to as *integrated enhancement*. The mean horizontal wind speed u is given in meter per second and the angle between the wind direction and aircraft flight directions is

denoted as $\varphi$ (in the following referred to as relative wind direction). Furthermore, it is assumed that no uptake by the soil takes place when the gas plume hits the ground and that the flight altitude is high enough (i.e. well above the Planetary Boundary Layer) to cover the entire vertical extent of the plume.

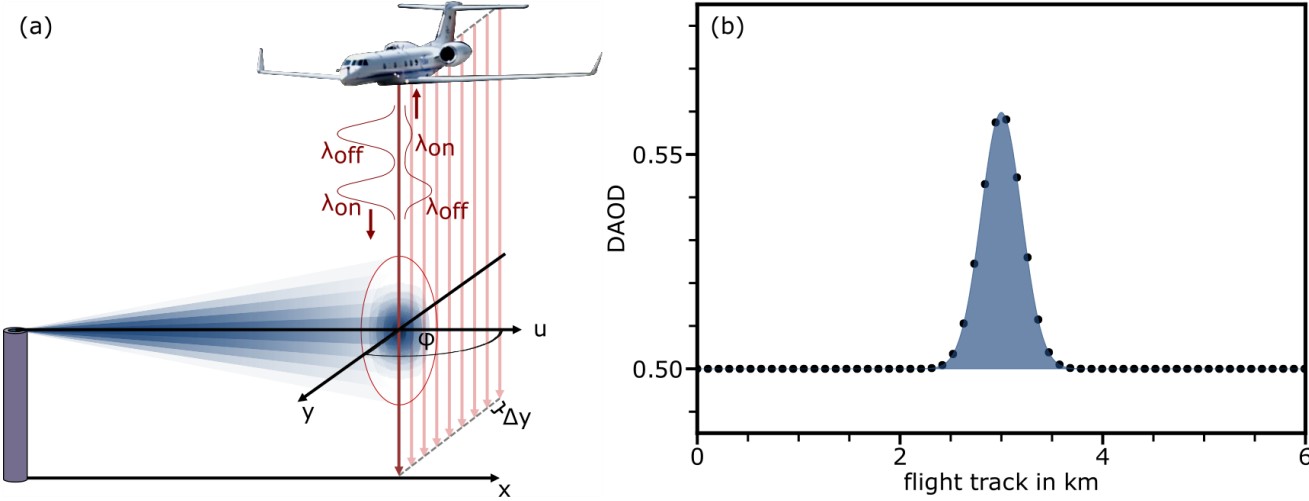

Figure 2: Crossing an exhaust plume illustrated by the example of CHARM–F carried on the aircraft HALO. (a) Two laser pulses are emitted towards the earth with short delay. The laser pulse with the wavelength $\lambda_{on}$ is on the absorption wavelength of $CO_2$, the laser pulse with the wavelength $\lambda_{off}$ is not. By comparing the backscattered intensities, the DAOD can be calculated (see Sect. 2.1). An ideal exhaust plume of a point source has a Gaussian-shaped mean concentration distribution both horizontally and vertically. (b) A perpendicular crossing of the plume yields a Gaussian-shaped DAOD dataset.





The two closely spaced sounding wavelengths are selected in such way that on the one hand the impact from unknown particles is minimized and on the other hand the absorption by water vapor is as low as possible. This is due to the very weak water vapor differential–absorption cross section, which is more than 4 orders of magnitude smaller than the differential–absorption cross section for $CO_2$. Thus the influence of additional water vapor in the plume released by the cooling or coal drying systems of the power plant is negligible (Kiemle et al., 2017). Moreover the selected $CO_2$ absorption line is sufficiently temperature-insensitive, such that the influence of temperature variations within the plume can be neglected (see also Kiemle et al. (2017)). Under these conditions, the flux error is mainly (Strandgren et al., 2020)driven by uncertainties of the four parameters A, $\Delta\sigma$, u, and $\varphi$. Assuming that these parameters are not correlated, the relative accuracy in the flux calculation can then be estimated by error propagation means:

$$\frac{\delta q}{q} = \sqrt{\left(\frac{\delta A}{A}\right)^2 + \left(\frac{\delta(\Delta\sigma)}{\Delta\sigma}\right)^2 + \left(\frac{\delta u}{u}\right)^2 + \left(\frac{\delta\varphi}{\tan(\varphi)}\right)^2} \quad (4)$$

With $\delta A/A$, $\delta(\Delta\sigma)/\Delta\sigma$, $\delta u/u$ denoting the relative uncertainties of these parameters. From this it is obvious that crossing the plume perpendicular to the wind direction as displayed in Fig. 2a would give the highest accuracy for any fluctuation of the wind direction $\delta\varphi$. On the other hand, atmospheric condition at low wind speeds or situation with high atmospheric turbulence are in general less favorable because of the high uncertainty in the mean wind speed and wind direction.

## 2.2    Background separation

For the calculation of the integrated enhancement A and its uncertainty it is crucial to distinguish between the DAOD value attributable to the background concentration of $CO_2$ and the fraction attributable to the exhaust plume of the point source. The measured DAOD along the flight track is the sum:

$$DAOD = DAOD_b + \Delta DAOD \quad (5)$$

Where $DAOD_b$ refers to the background term and $\Delta DAOD$ is the enhancement due to the plume interaction. A complicating factor is that the background term may not be constant. There are small variations in local $CO_2$ concentration from other anthropogenic sources (traffic, cities, etc.) or local interaction with the biosphere. Also, small $CO_2$ gradients caused by sounding of different air masses in the vicinity of the plume may have an impact on the background term. In the following, we describe a suitable method that enables us to extract $\Delta DAOD$ from the measured dataset.

An example of the plume extraction procedure is shown in Fig. 3. The plume must first be detected as an enhancement not attributable to noise in the data. For this we examine a 200 m running mean of the DAOD dataset (Fig. 3a). The larger the window for the running mean, the less noise is present and the clearer the plume enhancement can be seen. On the other hand, peaks threaten to be blurred if the window width becomes too large.



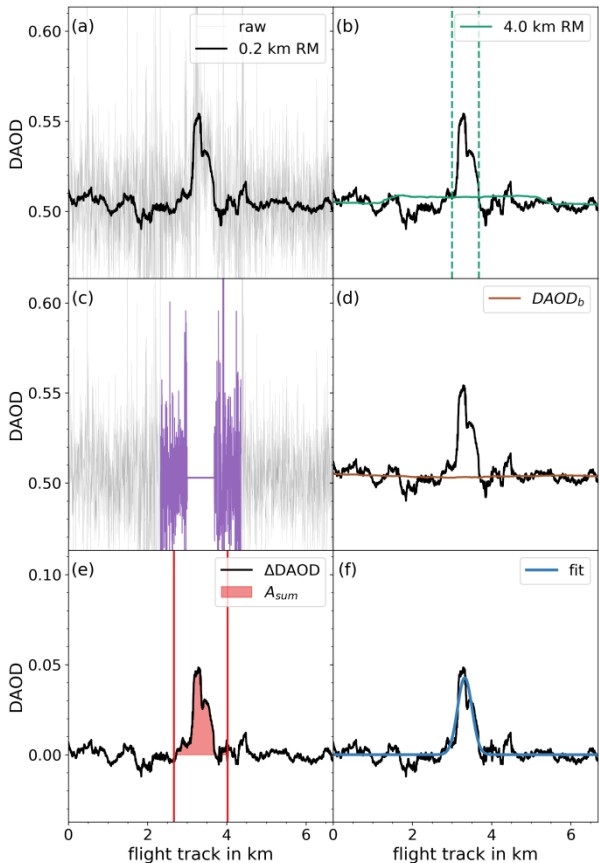

Figure 3: Plume crossing at a point source distance of 1.53 km. Colored in grey is the raw DAOD data. In (a) the gray curve shows the raw data, while the black curve shows a 0.2 km running mean (RM). In (b) the green curve is a 4 km running mean (RM). Green vertical dashed lines mark the intersections between the 0.2 km RM and 4 km RM, which are defined as the plume's limits. Colored purple in (c) shows the region of the data used to construct a mean value of the data before and after the plume's limits. This mean value is used to bypass the plume enhancement and is also colored purple. In (d) again a 4 km running mean over the bypassed dataset is shown in brown. This brown data is used as background term $DAOD_b$. Finally, in (e) and (f) the enhanced term $\Delta DAOD$, i.e. difference between 0.2 km RM and $DAOD_b$, is plotted in black. Note the different scale on the y-axis. In (e) the area underneath the curve is colored red, as an example of the parameter $A_{sum}$ determined with a Riemann sum. Alternatively, a Gaussian fit can be applied to $\Delta DAOD$, providing the parameter $A_{fit}$ as a fit-parameter, as shown as a blue line in (f).

Starting from the middle of the plume enhancement we define the plume's limits as the intersections between the 0.2 km running mean (RM) and another 4 km running mean (Fig. 3b). The choice of 0.2 km is made because it corresponds to the diameter of the pixels of the simulation (see Sect. 4). Hereby the magnitudes of the measured data can be compared with the simulated ones later. For the definition of the plume's limits a running mean with such a width must be used that the plume enhancement is blurred. Experience suggests that this is the case with a running mean width of 4 km, as shown in Fig. 3b. The limits are then defined as the intersections between the 0.2 km running mean and the 4 km running mean. Further on the data within the limits is replaced by the arithmetic mean of the data outside the plume. For the calculation of this mean we consider




a window with a width equal to that of the plume, colored violet in Fig. 3c. At last, we execute another 4 km running mean over the raw dataset, with bypassed plumes, resulting in the background term $DAOD_b$, shown in blue in Fig. 3d.

The mean wind speed u and its mean relative direction $\varphi$ are extracted from model data provided by ECMWF. The molecular mass M and the differential–absorption cross section $\Delta\sigma$ are physical properties of $CO_2$ and available in various databases such as HITRAN2016 (Gordon et al., 2017). The only parameter that results from a measurement by CHARM–F, or a respective simulation, is the integrated enhancement A. For this purpose Amediek et al. (2017) described two distinct methods. The first method is a Riemann sum over all enhancement values $\Delta DAOD_i$, multiplied with their respective spatial distance $\Delta y_i$ between

two successive data points:

$$A_{sum} = \left(\sum_i \Delta DAOD_i \cdot \Delta y_i \right) \tag{6}$$

The second method makes use of the fact that, on average, the plume is subject to Gaussian dispersion behavior. According to the function F(y) in Eq. (7), a nonlinear least squares fit is applied to the $\Delta DAOD$ values of the plume.

$$F(y) = \frac{A_{fit}}{A_2 \cdot \sqrt{2\pi}} \cdot e^{-\frac{1}{2} \cdot \left(\frac{y - A_1}{A_2}\right)^2} \tag{7}$$

By doing so the integrated enhancement is obtained as the fit parameter $A_{fit}$. $A_1$ is the peak's position along the flight track and $A_2$ the turbulent dispersion parameter, which is a measure for the width of the plume. The fit method yields very low values for the uncertainty of the parameter A. However, the Riemann sum is not depending on any model assumption for the calculation of the integrated $\Delta DAOD$ along the flight track. Both methods were investigated in the course of this work and showed nearly identical results. Therefore, the results in Table 1 correspond to the mean value of the two methods.

**3    Airborne measurements**

In this work the measurement flight of HALO on 23 May 2018 between 10:24 and 11:36 is investigated. Located in the south–east of the German federal state of Brandenburg, close to the Polish–German border, the *Lausitz Energie Kraftwerke AG* (LEAG) operates the coal–fired power plant Jänschwalde. It is one of the largest power plants in Europe, both in terms of annual electricity generation and annual $CO_2$ emissions. For the year 2017, the power plant operators have reported an emission

quantity of 24.0 Tg($CO_2$) to the European Environment Agency (E-PRTR, 2020). The exhaust gases of this power plant are emitted through the cooling towers at a height of ~ 120 m. Figure 4 shows the flight track of the aircraft, as well as a picture of the cooling towers. In total, the point source was flown over seven times downwind, two times upwind and once directly overhead the cooling towers. In three of the downwind overflights, no enhancement in DAOD is visible. For these transects the distance to the point source was greater than 4.6 km. At such distances it can be assumed that the exhaust gases are too

much diluted with the surrounding air to generate a measurable signal.


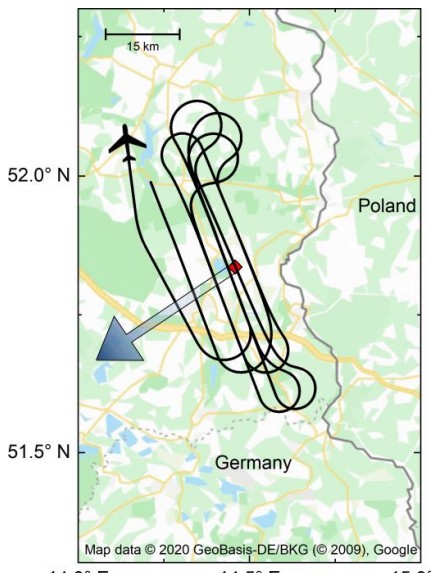
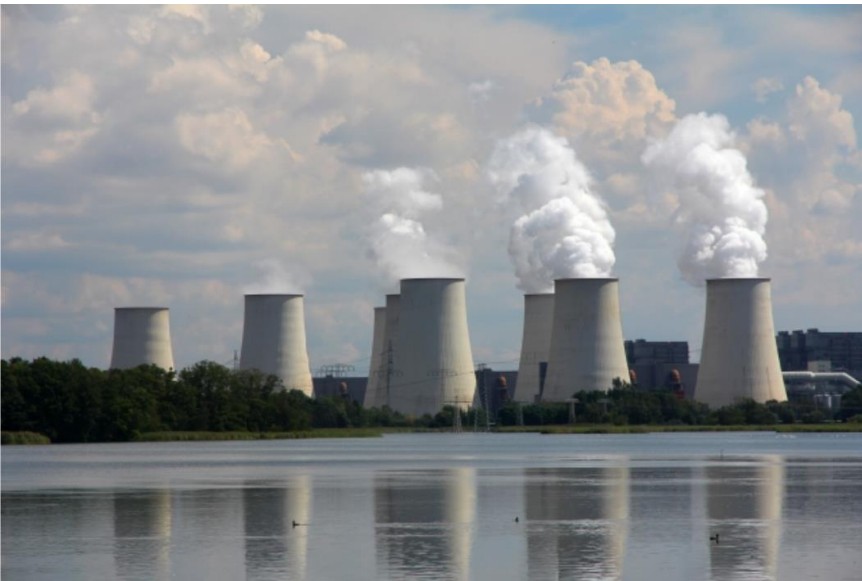

Figure 4: Flight track of HALO in the vicinity of the coal–fired power plant Jänschwalde. The black lines on the left depict the flight track of HALO between 10:24 and 11:36 on the 23 May 2018. The red square marks the position of the power plant Jänschwalde. The arrow shows the mean wind direction during the observation period. The right picture shows the nine cooling towers facing southwest. There the exhaust is released. The towers have a height of 120 m and horizontal spacing of 250 m and 50 m.

In order to evaluate the uncertainty of the calculated integrated enhancement, a total of 15 different data subsets centered to the plume's position, with varying width, are examined. To ensure that the Riemann sum completely covers the plume, the smallest subset is twice as wide as the plume limits, determined according to Fig. 3. The width of the remaining subsets is expanded by 400 m each.

To further evaluate Eq. (3) the differential–absorption cross section $\Delta\sigma$ is calculated using the Voigt-profile model with input from HITRAN 2016 data base for the line parameters (Gordon et al., 2017). This calculation requires the knowledge of pressure and temperature profiles, which are extracted from the simulation introduced in Sect. 4. For the lidar measurements, the online wavelength was tuned to the $CO_2$ absorption line center at $\lambda_{on}$ = 1572.02 nm, while the offline wavelength was adjusted to $\lambda_{off}$ = 1572.12 nm in the wing of this line (cf. Fig. 1b). Based on this wavelength selection and a flight altitude of 8000 m, the

background $DAOD_b$ is approximately 0.5, while the plume causes a ~ 10 % enhancement to this value (~ 0.05), as depicted in Fig. 3. The absorption cross section is not constant over the plume's vertical extension, mainly due to the decreasing pressure and resulting decrease in collisional line broadening with altitude. The relative change in absorption cross section along the vertical course of the plume depends on the exact online position with respect to the absorption line center. To take a possible cross section change for our measurements into account, representative mean values for the distances at $x_1$ = 1500 m and $x_2$ =

4700 m (see Table 1) were calculated using the slender plume approximation (Amediek et al., 2017; Seinfeld and Pandis, 1997):



$$\Delta\sigma(x_{1,2}) = \frac{\int_0^{max} \Delta\sigma(z) \cdot \left( e^{-\frac{(z-h)^2}{2\cdot\left(\sigma_z(x_{1,2})\right)^2}} + e^{-\frac{(z+h)^2}{2\cdot\left(\sigma_z(x_{1,2})\right)^2}} \right) dz}{\int_0^{max} \left( e^{-\frac{(z-h)^2}{2\cdot\left(\sigma_z(x_{1,2})\right)^2}} + e^{-\frac{(z+h)^2}{2\cdot\left(\sigma_z(x_{1,2})\right)^2}} \right) dz} \tag{8}$$

In this equation, the ground ($z = 0$ m) and max $= 4000$ m denote the integration boundaries. h is the height of the cooling

towers. The key parameter in this equation is the turbulence parameter $\sigma_z$, which is a proxy for the plume extension in the

vertical direction, at the respective distance. Different expressions for this parameter for various atmospheric stability

conditions can be found in the literature, e.g. Seinfeld and Pandis (1997).

Assuming a moderately turbulent atmosphere, we found plume widths of $\sigma_z = 170$ m and $600$ m for the two distances. However,

if the atmospheric turbulence is less pronounced, the vertical plume widths are only $\sigma_z = 90$ m and $250$ m, respectively. Due

to lack of further information on turbulence characteristics during our measurements, we consider both plume widths in the

calculation below. The change of $\Delta\sigma(z)$ versus altitude above ground in Eq. (9) is calculated at grid cell spacing of 1 m in the

vertical direction using the following 2$^{nd}$ order polynomial function:

$$\Delta\sigma(z) = 7.10652 \cdot 10^{-27} + 8.60755 \cdot 10^{-31} \cdot z + 8.02673 \cdot 10^{-35} \cdot z^2 \tag{9}$$

Consequently the differential-absorption cross section at the surface scattering elevation (i.e. 70 m a.s.l.) corresponds to $\Delta\sigma(z$

$= 0$ m) $= 7.10652 \cdot 10^{-27}$. The constant factors of this equation are the result of fitting this function to some representative cross

section values from Voigt-profile calculations over the altitude range of 4000 m. The deviations of this approximation to the

*exact* Voigt-profile calculations are less than 0.1 %, which is regarded negligible. Finally, Eq. (8) gives following results:

$\overline{\Delta\sigma(x_1 = 1500\ m)} = 7.27 \cdot 10^{-27}\ m^2 \pm 0.04 \cdot 10^{-27}\ m^2$ and $\overline{\Delta\sigma(x_2 = 4700\ m)} = 7.47 \cdot 10^{-27}\ m^2 \pm 0.24 \cdot 10^{-27}\ m^2$.

The overscore indicates the mean value of the aforementioned turbulence scenarios with corresponding plume vertical widths

at each distance and the errors indicate the differences. Close to the source (~ 1500 m), the relative cross section uncertainty

is ~ 0.6 % and therefore negligible, whereas at a distance of 4700 m, the relative error is ~ 3.2 % and not negligible in the

overall error budget outlined by Eq. (4).

Possible systematic errors, due to uncertainties of the line parameters, are less than 2 % (Gordon et al., 2017). Errors according

to the wavelength setting with the CHARM–F instrument are considered very small compared to the other contributors and

therefore need not be extensively discussed in this study (~ 0.5 %, see Amediek et al. (2017)).

The wind data are taken from operational analysis data of the ECMWF model. This is done by first interpolating the 4-

dimensional gridded model data onto the flight path at the altitude of the power plant's exhaust shaft. Secondly, a mean value

of the wind speed and direction along the flight track, as well as an estimate of their relative errors is calculated according to

Ackermann (1983).





Table 1: Flight measurement results of individual crossings for the Jänschwalde power plant on 23 May 2018, following the nomenclature of Eq. (3)

| Crossing | | | Measurement | | | | | |
|---|---|---|---|---|---|---|---|---|
| local time | path in km | distance in km | mean q in kg($CO_2$)/s | q in kg($CO_2$)/s | A in m | $\Delta\sigma$ in $10^{-27}$ m$^2$ | mean u in m/s | mean $\varphi$ in ° |
| 10:50 | 200 | 1.46 | | $760 \pm 60$ | $15.36 \pm 0.67$ | $7.27 \pm 0.04$ | | |
| 10:57 | 268 | 4.77 | | $470 \pm 40$ | $9.04 \pm 0.42$ | $7.47 \pm 0.24$ | | |
| 11:10 | 388 | 1.67 | $650 \pm 240$ | $950 \pm 80$ | $19.29 \pm 0.46$ | $7.27 \pm 0.04$ | $5.06 \pm 0.36$ | $103.34 \pm 6.40$ |
| 11:27 | 536 | 1.78 | | $420 \pm 40$ | $8.45 \pm 1.11$ | $7.27 \pm 0.04$ | | |

Table 1 shows the measured integrated enhancements, the wind data and the resulting fluxes for the four exploitable
overflights, as well as the obtained mean values, under the assumption that during the measurement both the wind direction, as well as the wind speed are reasonably constant. The flight segments were not exactly perpendicular to the mean wind direction. With a relative angle of $\varphi = 103°$ a correction factor of $\sin(103°) = 0.97$ was applied (see Eq. (3)).

The individual flux uncertainties, calculated with Eq. (4), are relatively small and range between 8–10 %. It is to be emphasized that the integrated enhancement A is the only parameter in the calculation of the instantaneous flux in Eq. (3), coming from
the IPDA lidar measurement itself. On average, two-tenth of this individual measurement uncertainty is due to the uncertainty of the integrated enhancement $\delta A/A$. Taken together, one-tenth can be attributed to the uncertainty of the differential–absorption cross section of $CO_2$ $\delta(\Delta\sigma)/\Delta\sigma$ and the mean relative wind direction $\delta\varphi/\tan(\varphi)$. The major contributor to the flux uncertainty, however, is the uncertainty of the mean wind speed $\delta u/u$, which accounts for two-thirds.

The reported value of 760 kg($CO_2$)/s (24.0 Tg($CO_2$)/yr) lies within the error range of the mean value of $650 \pm 240$ kg($CO_2$)/s
($20.3 \pm 7.9$ Tg($CO_2$)/yr). Nevertheless, the variations between the individual crossings are very large, both in the integrated enhancement A, as well as in the calculated fluxes. The second and third crossings differ by approximately a factor of 2 (see Table 1). These variations cannot be explained by our uncertainty estimation, but rather by atmospheric turbulence that distorts the plume. This work therefore further investigates the influence of atmospheric turbulence and the resulting inhomogeneity in the propagation of exhaust plumes. To achieve this, we make use of the mesoscale numerical weather prediction system
model WRF (Weather Research and Forecasting model).

## 4    Simulation setup

To investigate the influence of atmospheric turbulence and the resulting inhomogeneity in the propagation of exhaust plumes, we use WRF-ARW, the Advanced Research Version of the Weather Research and Forecasting model (Skamarock et al., 2008).



It is a well-established platform to investigate the transport of plumes (Zhao et al., 2019; Bhimireddy and Bhaganagar, 2018;
Yver et al., 2013). The model configuration can be found in Table 2.

Table 2: WRF model configuration

|  | **Setting** | **Reference** |
|---|---|---|
| **WRF version** | WRF 3.8.1 | Skamarock et al. (2008) |
| **Dynamical solvers** | Advanced Research WRF | |
| **Meteorological boundary conditions** | Operational ECMWF analysis | ECMWF (2018) |
| **Simulated time span** | 06:00 UTC on 21 June – 06:00 UTC on 24 June in 2018 | |
| **Spin-up** | 6 h | |
| **Number of vertical layers** | 56 | |
| **Model top** | 200 hPa | |
| **Radiation** | Rapid Radiative Transfer Model Scheme (ra_lw_physics = ra_sw_physics = 4) | Iacono et al. (2008) |
| **Microphysics** | Morrison 2-moment Scheme (mp_physics = 10) | Morrison et al. (2009) |
| **Land surface model** | Unified Noah Land-Surface Model (sf_surface_physics = 2) | Tewari et al. (2004) |
| **Surface layer physics** | Revised MM5 Scheme (sf_sfclay_physics = 1) | Jimenez et al. (2012) |

Considering typical source distances of the measurement crossings (see Table 1), as well as the spread of the plumes (see Fig. A1), it is clear that our investigations need to be implemented with a horizontal resolution in the sub–kilometer range. To achieve this, we introduce three nested domains, with the coordinates of the middle cooling tower as center of the domains (see Fig. 5). The domain configurations can be found in Table 3. As meteorological initial and boundary conditions, operational

ECMWF analysis data is used with a horizontal resolution of 9 km.





Table 3: Configuration of quadratic domains

| Domain | D1 | D2 | D3 |
|---|---|---|---|
| **Horizontal resolution** | 5 km | 1 km | 0.2 km |
| **Computational time step** | 30 s | 5 s | 1 s |
| **Number of grid points** | 100 | 150 | 175 |
| **Domain size, W-E and S-N** | 500 km | 150 km | 35 km |
| **Planetary Boundary Layer Physics** | MYNN Level 2.5 Nakanishi and Niino (2009) | MYNN Level 2.5 Nakanishi and Niino (2009) | LES PBL Moeng et al. (2007) |
| **Eddy coefficient option** | 2d Deformation (km_opt=4) | 2d Deformation (km_opt=4) | 3d TKE (km_opt=2) |
| **Turbulence and mixing option** | Simple diffusion (diff_opt=1) | Simple diffusion (diff_opt=1) | Full diffusion (diff_opt=2) |

As suggested by Powers et al. (2017) we run the inner domain D3 as a large eddy simulation (WRF-LES). This makes it possible to resolve local turbulence (Moeng et al., 2007). Several studies show that WRF-LES is an adequate tool to model plume trajectories, as well as turbulence and passive tracer dispersion (Nunalee et al., 2014; Nottrott et al., 2014).

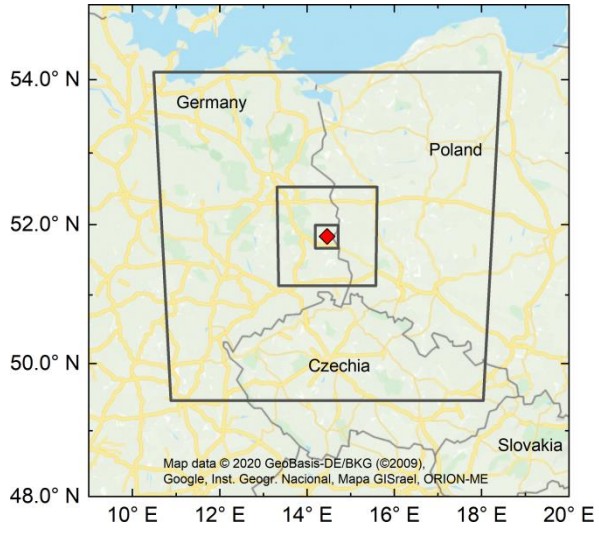





Figure 5: Location of three times nested domains (black squares) of the WRF simulation. They are centered with respect to the power plant Jänschwalde (red square). The domains have a side length of 500 km (D1), 150 km (D2), 35 km (D3) and a horizontal resolution of 5 km (D1), 1 km (D2) and 0.2 km (D3). Vertically, 57 eta levels are introduced ranking from the ground up to a top layer pressure of 200 hPa.

Only the plume of the power plant is simulated, without any $CO_2$ background field. WRF-ARW has the option to predefine a

tracer variable tr(t,x,y,z) which has the properties of a passive tracer, as used in Blaylock et al. (2017). It represents a four-dimensional field of space-time. A detailed description of the calculation of simulated DAOD can be found in Appendix A. Therein Eq. (A3) is used to calculate the DAOD enhancement corresponding to the horizontal dispersion of the tracer, as shown in Fig. 6.

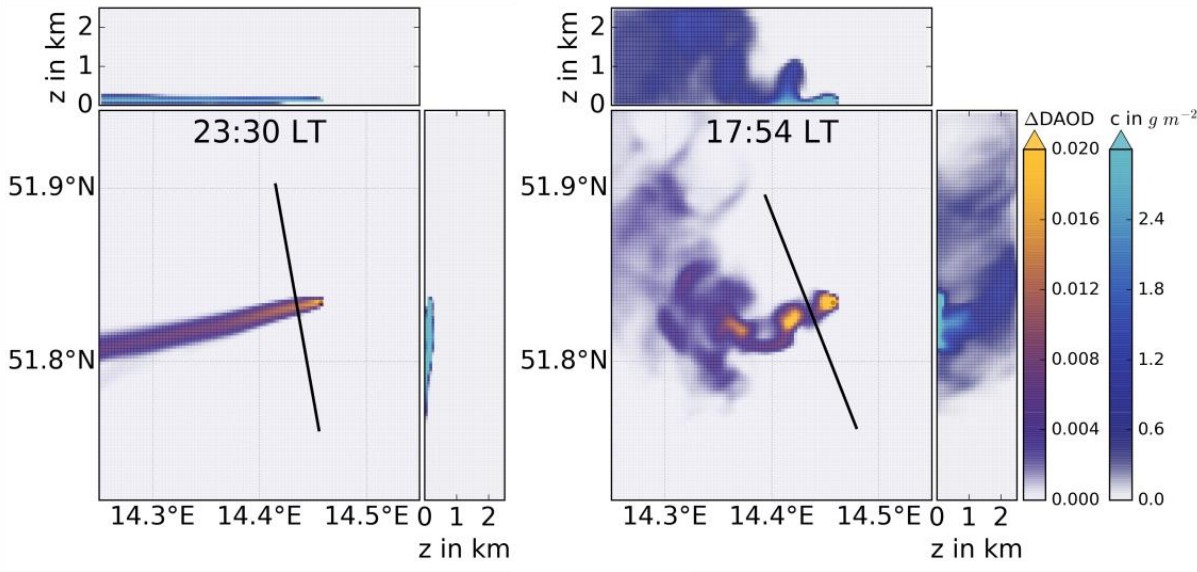

Figure 6: Exemplary snapshots of simulated exhaust plume. The flight track of the virtual plume overflight is shown as a black line. The first colorbar represents the DAOD enhancement and refers to the respective middle panel, which shows the horizontal dispersion of the plume. The second colorbar represents the mass per area and refers to the top and right panels, which show the vertical dispersion. For the DAOD, enhancement values beneath 0.008 cannot be distinguished from noise and are display blue. Values higher than 0.01 exceed the noise and can be identified as plume enhancement in a real measurement. A ΔDAOD value of 0.02 corresponds to an enhancement of 4 % with respect to a background of 0.5 (cf. Fig. 3). The color maps follow the guidelines for a perception-based color map presented by Stauffer et al. (2015).

The WRF simulation provides a data output every 2 minutes. One virtual plume crossing is evaluated for each output time step

at a point source distance of 1.5 km. This corresponds to our measurements (see Table 1). Since neither background field nor noise is simulated, it does not matter at which distance to the point source the virtual flyover takes place. Nevertheless, we try to match the virtual survey as closely as possible to real conditions. Just as in the real measurement, the virtual crossings are arranged perpendicular to the propagation direction of the plume (cf. Sect. 2.1). However, in a turbulent atmosphere it is not trivial to precisely identify this direction of propagation. In this work we consider the center of mass of the emitted tracers





within a radius of twice the point source distance, i.e. 3 km. A connecting line between this center of mass and the point source

corresponds to the propagation direction.

For the calculation of the virtually retrieved emission rate the mean wind speed and direction are needed (see Eq. 3). To obtain

these from the simulation, the following procedure is performed. First, for each data output step the horizontal wind

components at the mean height of the plume are retrieved by a vertical integral, weighted by tracer mass content. Second, the

resulting 2D wind field is linearly interpolated onto the virtual flight path, yielding a 1D field with the horizontal wind

components along the flight track. Last, the wind components are integrated, weighted with the DAOD along the flight track,

resulting in the mean wind used for calculation.

## 5    Simulation results

WRF is able to simulate realistic plume dispersion. The DAOD enhancement values correspond to our measurements.

Exemplary snapshots of the simulated plume during the course of a whole day can be found in the Appendix A in Fig. A2.

Additionally, an animated GIF of the simulated plume can be found under https://doi.org/10.5281/zenodo.4266513 (Wolff,

2020). In the nocturnal absence of solar irradiation the turbulence decreases, leading to narrow, homogeneous plume

dispersion, within a laminar flow. The exhaust plume follows Gaussian behavior, as depicted in Fig. 6 at 23:30. Contrary to

this, we find boundary layer turbulence during daytime.

Strong solar heating of the surface generates convective air masses, which in turn cause a cascade of eddies. Consequently,

locally reverse and counter-gradient flow, i.e. flow opposite to the main wind direction, emerges. This results in local *puffs* of

above-average column concentration enhancements within the exhaust plume, while eddy-generated local flow in the same

direction as the ambient wind causes *constrictions* of lower column concentrations in a plume (Stull, 1988). Such plume

structures deviate from Gaussian behavior, as can be seen in Fig. 6 at 17:54.

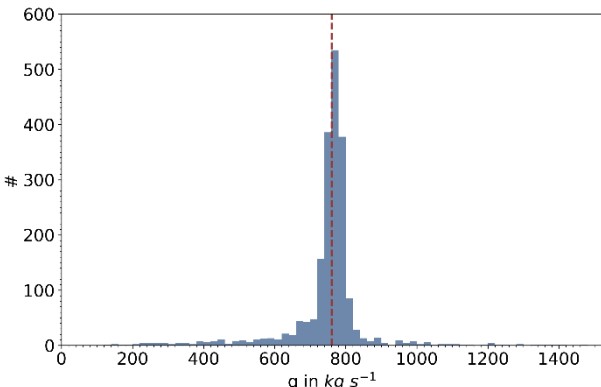

Figure 7: Histogram of virtually retrieved emission rates. The histogram shows a slightly skewed distribution towards smaller emission
rates than the input emission rate (red dashed line). It depicts all 1980 emission rates retrieved over 66 simulated hours between 12 UTC
on 21 May and 6 UTC on 24 May.

Locally increased $CO_2$ column concentration results in a high value in the integrated enhancement A, in contrast to an overflight over a constriction. Following Eq. (3) this corresponds to a high value of the emission rate q. It should also be stressed that the spatial extent of such puffs is smaller than that of complementary constrictions. Therefore, a skewed distribution of the retrieved emission rates is to be expected, as Fig. 7 confirms.

On 23 May 2018, four measurement flyovers of the power plant Jänschwalde took approximately one hour, as presented in

Sect. 3. As spin-up we discard the first 6 hours of the simulation (see Table 2). That is 66 hours of simulation, which leaves us with a total of 1980 virtual plume flyovers. The corresponding results of the emission rate, which has been calculated using Eq. 3, are displayed as a histogram in Fig. 7 and as a time series in Fig. 8a.

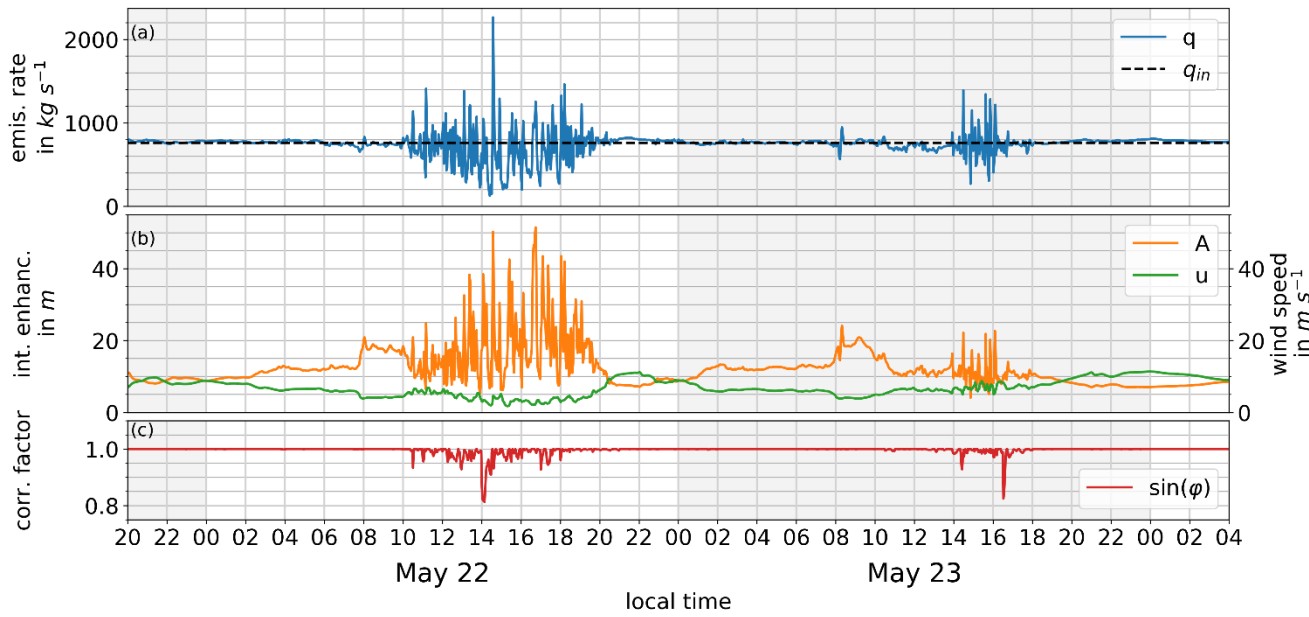

Figure 8: Virtual overflight results in the course of the day. Midday turbulence causes a deviation of the retrieved emission rates q from the input emission rate $q_{in}$ in (a). This can correspondingly be seen in the integrated enhancement A in (b), while the wind speed u shows comparatively small variations. It is during the midday turbulence, that the virtual flight track is not exactly perpendicular to the instantaneous wind direction at the plume crossing, which is shown in the correction factor $\sin(\varphi)$ in (c). In the night hours, as well as the morning the retrieved emission rates agree very well with the input emission rate $q_{in}$.

In Figure 8a it can clearly be seen how the diurnal course of solar irradiation influences the retrieved emission rates q. The random occurrence of inhomogeneities in the plume propagation leads to large variations in the results of successive crossings.

These large variations can be reduced by averaging over a multitude of retrieved emission rates.

Next, we investigate how often the exhaust plume must be surveyed in order to achieve a mean emission rate with satisfactory accuracy. From experience with the Jänschwalde measurement presented in Sect. 3, as well as other point source measurements during the CoMet campaign, which are not presented in this work, we assume a time delay in the range of 6 to 18 minutes





between two successive crossings. With typical wind speeds in the range of 5–8 m/s and spatial scales of *puffs* and *constrictions*

of about 1–2 km, our range of time delay exceeds the residence time of coherent plume structures, thus preventing repeated

measurements of identical air masses. The model setup provides one measurement every 2 minutes, resulting in a vast number

of permutations of successive virtual crossings available for merging (see Table A1). For each of these permutations, a mean

value is calculated, which is then compared with the initiated emission rate $q_{in}$. In order to evaluate the turbulence-induced

inhomogeneity in the daily course, we compare two-hour time frames. We execute a total of 60 virtual overflights in such two-

hour time frame. The number of possible permutations increases exponentially to 5000 if four crossings are merged and even

on to 312500 if seven crossings are merged (see Table A1). This high number of permutations is based on the identical 60

single crossing emission rates, which are displayed in the histograms in Fig.9.

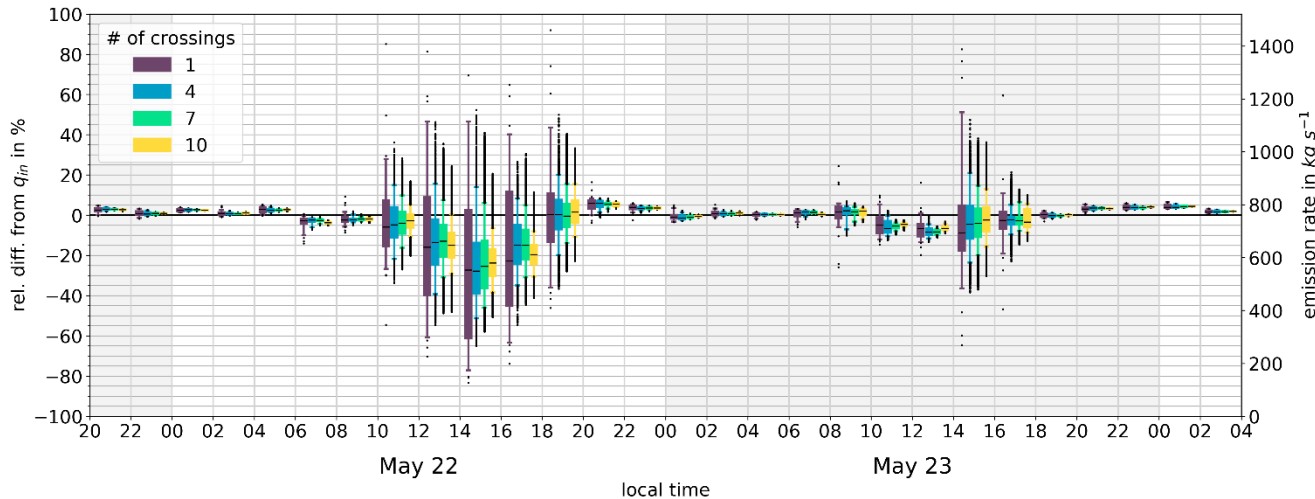

Figure 9: Box-whisker-plot of the relative difference to the input emission rate $q_{in}$ within two-hour time frames. The right axis shows the associated retrieved emission rates. The width of the distribution decreases with higher number of crossings. For all time frames it can be stated that with increasing number of merged crossings the width of the distribution decreases. The largest differences to $q_{in}$ are observed in the afternoon. Different colors represent a different number of virtual crossings merged for averaging. The inner boxes range from the first to the third quartile, thus containing 50 % of the values. The median is marked within as a black dash. The upper whisker is drawn up to the 95[th] percentile, while the lower whisker is drawn to the 5[th] percentile Consequently 90% of the values are in between the two whiskers. All values outside the whiskers are outliers and plotted as dots.

Figure 9 presents the resulting distribution of this relative difference to the input emission rate as a box-whisker-plot. The

spread of respective box-whisker-plot is an indicator of turbulence. It is evident that the spread of the relative differences

decreases, and the measurement precision increases, with a higher number of overflights that are merged for averaging. A high

emission rate measured by a single overflight scanning a puff is compensated if the subsequent overflight measures a lower

emission rate. With a higher number of overflights averaged, it is more likely to measure both high and low concentration air

masses. Yet, although the precision can be improved with increasing overflight number, even for ten overflights it is inferior

to the precision of night time measurements. Additionally, not only the precision, but also the accuracy is compromised during





times of strong turbulence, i.e. in the afternoon. As aforementioned, the spatial extent of turbulence induced puffs is smaller than the one of the complementary constrictions. Therefore, such puffs are likely to be less frequent and only partially scanned when measured at a low sampling frequency. Consequently, the retrieved emission rates will be biased low. This is an effect that occurs especially during strong turbulence. In Fig. 9 a strongly turbulent day (22 May) is compared to a less turbulent day (23 May). Both precision and accuracy are superior on a less turbulent day.

In contrast, the night hours show little turbulence, and high precision. Even with a single overflight, small differences to the true emission rate are to be expected. Here, a higher number of overflights will only cause minor improvements. At this point, it should be mentioned that the representation of nightly plume propagation must be critically reviewed. The plume height decreases so much that the propagation takes place only in lowest four model layers. The fact that a bias of approx. ±5% remains at night is not surprising from this point of view. This study should therefore be understood as a qualitative assessment.

The key finding is that avoiding midday turbulence brings an enormous improvement for both precision and accuracy. Even with a significantly higher number of measurement overflights, a comparable improvement cannot be attained. For future campaign planning, we therefore recommend to also perform measurements at night or in the morning, which is possible with an IPDA lidar system.

## 6  Discussion

Regarding the lidar measurements during the CoMet campaign on 23 May 2018, we find that the mean emission rate, derived from repeated plume overflights, is in rough agreement with the average emission reported by the power plant operator for the year 2017. The cross-sectional flux method is straight forward to apply. The exhaust plume generates column enhancements in the differential absorption optical depth (DAOD), with a good signal-to-noise ratio, in the order of 10 %. The product of enhanced column concentration integrated along the flight track and mean wind speed, provides the flux through the lidar

cross-section, at the instant of the overflight. This instantaneous flux of an individual overflight measurement can be determined with an error ranging between 8–10 %. This error is mainly driven by uncertainties of the integrated enhancement, the differential-absorption cross section, the mean relative wind direction, and the mean horizontal wind speed. On average, we find that two-tenth of the flux error can be attributed to uncertainty in the determination of the integrated enhancement, i.e. the integrated enhancement of the DAOD signal, which is the only parameter that needs to be derived from the IPDA lidar

measurement. One-tenth can be attributed to the uncertainties of the differential-absorption cross section of $CO_2$ and the relative wind direction, taken together. The main source of error, however, is the mean horizontal wind speed, with a contribution of two-thirds. This highlights the need for more accurate wind information.

It is necessary to distinguish between instantaneously measured flux and actual emission rate. In theory, an exhaust plume behaves Gaussian on average and the mean emission rate of the point source lies within the error range of the instantaneously

measured fluxes. Contrary to this, our overflights reveal large variations between the individually retrieved instantaneous fluxes, which cannot be attributed to measurement uncertainties. These variations do not occur because the measurement error





increases, but because plume segments with varying $CO_2$ content are probed. The actual measurement error is minor compared to these variations (cf. Table 1). As described in Sect. 5, strong solar heating causes turbulence, which forces the plume to deviate from Gaussian behavior. This deviation can be restricted by averaging over a multiple of instantaneous fluxes, as the
results from our measurement flights suggest.

To analyze this effect in more detail, we employ the atmospheric transport model WRF in a high-resolution large eddy simulation (LES) setup. We find that the model simulates realistic plume dispersion. Typical DAOD values, as well as turbulence induced distortions, show the same order of magnitude as our measurements. However, as we evaluate only four overflights in the measurement, we cannot make any statement about the absolute accuracy of the simulation, which is also
not the intention of this work. Qualitatively, the simulation provides the following insights. During the night the plumes are weakly distorted and have Gaussian shape, because laminar flow dominates. Over the course of the day, turbulence increases, reaching its peak in the mid-afternoon and distorting the plumes to non-Gaussian shapes. Thus, with increasing turbulence, a higher number of crossings are required for averaging in order to obtain sufficient emission rate precision. According to our simulation, nighttime measurements require fewer overflights. Under such conditions even a single instantaneous cross-
sectional flux measurement yields high accuracy. In cases of very pronounced turbulence (i.e. in the afternoon) even an impractical high number of overflights will neither reach the precision, nor the accuracy of a single nighttime overflight. The determination of such conditions makes atmospheric transport modelling an indispensable aid for the presented and upcoming measurements.

## 7    Conclusion and outlook

The present study continues the investigations by Amediek et al. (2017) on the quantification of fluxes of local greenhouse gas emission sources using the integrated-path differential-absorption (IPDA) lidar CHARM–F and the cross-sectional flux method. While the preceding study was concentrated on $CH_4$ emissions from hard coal mines, we exploit here the results from the CoMet campaign in 2018. We investigate $CO_2$ plume overflights of the coal-fired power plant Jänschwalde, conducted to quantify its emission rate and to assess how accurately the cross-sectional flux method can be applied. Since CHARM–F
measures both greenhouse gases simultaneously, our findings also apply to isolated $CH_4$ point sources.

With regard to cross-sectional flux measurements, the current work suggests avoiding mid-afternoon periods of strong turbulence. On the one hand this is because the uncertainties in the wind field are most pronounced at these times, being a major source of error in a single measurement. On the other hand, this is due to the distortions of exhaust plumes in a turbulent wind field, which lead to substantial deviation from Gaussian plume dispersion. Under strong turbulence, the cross-sectional
flux method cannot provide an accurate measurement of the emission rate, not even in the average of a vast number of overflights. Therefore, measurement flights performed during nighttime are preferable. In this respect, intrinsic independence from solar irradiation is a clear advantage of active remote sensing over passive approaches. Whenever sunlight is needed to perform the measurement, less turbulent conditions, for example in the morning after sunrise, or winter, should be preferred.



Further, it shall be pointed out that, with a lidar, cross-sectional plume measurements can also be performed over water bodies,
whose detrimental reflective properties often impede the use of passive remote sensing (Gerilowski et al., 2015; Larsen and Stamnes, 2006; Krautwurst et al., 2020). Therefore, plumes from offshore installations can also be addressed with this approach.

Independent of the location of the point source, there are restrictions regarding the adequate distance of a plume overflight to the point source. On the one hand, we report that at a point source distance of more than 4.6 km no enhancement is visible and
therefore no plume detection can be performed. On the other hand, we find that the uncertainty of the differential-absorption cross section increases with a larger vertical extension of the plume, which correlates with distance. At a point source distance of 1.5 km this uncertainty is negligible. With respect to the detectability of the plume, we are able to locate distinct enhancements at a distance of 1.5 km. Nevertheless, the closer to the point source the overflight takes place, the more constrained the plume and consequently the more pronounced the column enhancement is. It must be considered, however,
that the horizontal extension is also smaller and thus fewer data points lie within the plume. In case of CHARM-F this can be compensated by a higher repetition rate.

Apart from the CHARM-F measurements, the CoMet campaign also saw the deployment of other airborne instruments to measure atmospheric $CH_4$ and $CO_2$, supported by a variety of ground–based, in situ and remote sensing instruments. They were predominantly based in the vicinity of one of the major hot–spot regions of $CH_4$ emissions in Europe, the Upper Silesian
Coal Basin (USCB). Investigations of local and regional $CH_4$ emissions from this region are, in view of the preparation for the upcoming MERLIN mission, a particular field of interest. The possibility to synergistically use active remote sensing (lidar), passive spectrometry, and in situ measurements supported by modelling activities, allow for unique cross comparisons, which are beyond the scope of the present paper. Such cross comparisons will be subject of subsequent investigations, as well as other HALO measurement flights, as it flew along latitudinal trajectories, performed regional survey flights (e.g over the
Mount Etna) and also probed the local plume of not only Jänschwalde, but also Bełchatów in Poland, which is considered Europe's largest coal-fired power plant, in terms of $CO_2$ emission. The measured data can make an important contribution to the validation of existing satellite missions (e.g. Sentinel–5P, GOSAT). Further aircraft campaigns (e.g., CoMet–2.0) are foreseen which will provide additional opportunities for methodical refinements, including advancements on model-measurement synergies.

*Data availability.*

The data are available from the author upon request and will be made publicly available through the HALO Data Base www.halo.dlr.de/halo-db/ in due time.





# Appendix A

*Figures of individual HALO crossings on 23 May 2018:*





Figure A1: Individual transects listed in Table 1. Red vertical lines mark the smallest data extract used for the Riemann sum, as described in Sect. 3.


*Calculation of simulated DAOD:*

At the initialization position of the power plant, the value of the tracer variable is increased by the value 1 at each time step. In this study, it is defined only in the inner domain D3.

$$tr(t + \Delta t, x, y, z) = tr(t, x, y, z) + 1 \tag{A1}$$

Here, $\Delta t$ corresponds to the computational time step of the third domain, i.e. one second (Table 3). At the same time the tracer is distributed in the domain D3 by advection and turbulent dispersion. The corresponding mass concentration c(t,x,y,z) at any grid point x,y,z at time t is obtained as follows:

$$c(t, x, y, z) = tr(t, x, y, z) \cdot \frac{q_{in} \cdot \Delta t}{\Delta x \cdot \Delta y \cdot \Delta z(t, x, y, z)} \tag{A2}$$

For the input emission rate $q_{in}$ a constant value of 760 kg($CO_2$)/s (24.0 Tg($CO_2$)/yr) is initialized, which corresponds to the
total annual emission for the year 2017 reported to the European Environment Agency by the operators (E-PRTR, 2020). $\Delta x$ and $\Delta y$ correspond to the temporally and spatially constant horizontal size of a grid point (0.2 km). The vertical layer size $\Delta z(t,x,y,z)$ corresponds to the spatial distance between two model levels. In the simulation this distance is computed in pressure coordinates and depends on all four dimensions. Since the pressure varies only slightly between successive time steps, the temporal dependence of $\Delta z$ is small. At locations with flat topography the dependence of $\Delta z$ on the horizontal coordinates x
and y is also small, at locations with large topographic changes (e.g. steep slopes) the dependence is more significant. The product $\Delta x \cdot \Delta y \cdot \Delta z(t,x,y,z)$ corresponds to the volume of the respective grid box. Within this volume the value of the tracer variable and thus the concentration is constant.

In order to compare the simulated data with an IPDA lidar measurement, the concentration array must be summed up vertically and multiplied by the quotient of the differential-absorption cross section and the molecular mass:

$$DAOD_{wrf}(t, x, y) = \frac{\Delta\sigma}{M} \cdot \sum_{j=1}^{j_{top}} c(t, x, y, z_j) \cdot \Delta z_j(t, x, y, z) \tag{A3}$$

The index j marks the respective vertical layer. Consequently $j \in \{1,56\}$ applies, and $z_j$ is defined to correspond to the lower edge of the respective layer.

*Exemplary snapshots of simulated DAOD:*





Figure A2: Exemplary snapshots of simulated plume and virtual flight track. The daily solar irradiation causes a deep, convective boundary layer with turbulent plume dispersion within. In the nocturnal absence of solar irradiation, the boundary layer shrinks, leading to narrow, homogeneous plume dispersion, within a laminar flow. Every two minutes a virtual measurement is performed yielding 60 measurements within a time frame of 2 hours. One representative snapshot within the two-hour time frame is shown.




Table A1: Number of possible permutations of successive virtual crossings used for averaging

| # of measurements | # of possible permutations |
|---|---|
| 1 | 60 |
| 4 | 5000 |
| 7 | 312500 |
| 10 | 9765625 |

*Author contribution.*

SW analyzed the measurement data, performed the simulation and wrote most of the manuscript. AF, CK and GE supervised the measurement data analysis, as well as the simulation post processing. AA, AF, MQ and MW developed the lidar system and operated it during CoMet. AF was principal investigator of the CoMet mission. AA, AF, CK and GE designed the
measurement flights. AF, CK, GE and MQ contributed to the manuscript's text and figures.

*Competing interests.*

The authors declare that they have no conflict of interest.

*Acknowledgements.*

We thank Bastian Kern, Sabrina Arnold (both DLR Institut für Physik der Atmosphäre, Oberpfaffenhofen, Germany), and
Friedemann Reum (SRON Netherlands Institute for Space Research, Utrecht, Netherlands) for the helpful comments on a previous version of the paper. We further gratefully acknowledge Michał Gałkowski (Max Planck Institute for Biogeochemistry, Jena, Germany), Johannes Wagner (DLR Institut für Physik der Atmosphäre, Oberpfaffenhofen, Germany, now at Deutsches Patent- und Markenamt, Munich, Germany) and Andreas Luther (DLR Institut für Physik der Atmosphäre, Oberpfaffenhofen, Germany) for advising the simulation setup. We would like to thank DLR flight experiments for excellent
operations during CoMet.

*Financial support.*

We acknowledge financial support by BMBF (German Federal Ministry of Education and Research) through its AIRSPACE project (grants no. 01LK1701A/B/C), the German Science Foundation (Deutsche Forschungsgemeinschaft, DFG) within DFG





Priority Program SP 1294 "Atmospheric and Earth System Research with the Research Aircraft HALO (High Altitude and

Long Range Research Aircraft)". The HALO flights also received support in the frame of DLR project "KliSAW" and from

the Max Planck Society.

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
