# Peer review of "Determination of the Emission Rates of CO2 Point Sources with Airborne Lidar"

_Atmospheric Measurement Techniques, 2020_

## Referee Comment (RC1) · Anonymous Referee #1 · 25 Nov 2020

Manuscript "Determination of the Emission Rates of CO2 Point Sources with Airborne Lidar" from Wolff et al., reports on integrated-path differential-absorption lidar CHARM–F aircraft observations during the Carbon Dioxide and Methane mission (CoMet) campaign in spring 2018. They describe the measurement system and how these measurements have been processed to obtain differential optical absorption depth (DAOD) used to quantify CO2 emission rates of the coal-fired power plant Jänschwalde in Germany. They concluded that their results agree reasonable well with reported annual emission rates. However, they also found significant variations between individual overflights up to a factor of 2. They suspected that these variations are mostly driven by turbulence and performed high–resolution large eddy simulations to investigate this.

The manuscript is very well written and covers a topic relevant for Atmos. Meas. Tech.

[Figure]

I recommend publication after the minor issues listed below have been considered by the authors.

Abstract, line 8, sentence "Due to the large number of point sources and their global spatial distribution, a mobile measurement approach with fast spatial coverage is needed": Unclear, why a "mobile measurement approach with fast spatial coverage is needed"? Why mobile? Why spatially fast? Regular on-site monitoring could be even better. Please consider to revise this sentence.

Abstract, line 15, sentence "emission rates can be derived in terms of the cross–sectional flux method". I recommend to replace "in terms of" by "via" or equivalent.

Page 2, line 37, sentence "However, at the moment no operating satellite mission is able to reliably quantify emissions from large power plants." Unclear what exactly is meant here taking into account publications such as Nassar et al., 2017, and Reuter et al., 2019, which need to be cited (details see below in "References").

Page 2, line 60, sentence "Albedo variations basically influence the measurement precision (statistical uncertainty), not the bias.": This is a bold statement. Is it really true that the impact of albedo variation on the bias is zero or are corresponding biases only expected to be very small?

Page 3, line 67, sentence "This principle has been applied for . . .". The cross-sectional flux method has also been applied to satellite data, see Reuter et al., 2019. Please add this missing information.

Page 2, eq (2): Reader may wonder why delta sigma, the absorption cross section difference, does not depend on altitude. Later this aspect is addressed but I recommend to add some details on this aspect when presenting Eq 2.

Page 6, line 129: Reference to Strandgren et al., 2020: Unclear to which statement or fact the reference refers to. Please also add space between "2020)driven".

Page 7, line 154, sentence "For the definition of the plume's limits a running mean with

such a width must be used that the plume enhancement is blurred." Not mandatory but please consider to improve this sentence.

Page 7, line 156, sentence: "Further on the data within the limits . . .". Please improve this sentence.

Page 9, Figure 4, right (photo): I wonder if the authors have the appropriate right to use the photo. Please confirm.

Page 10, line 213: What is "surface scattering elevation"? Just the surface elevation of the ground scene?

Page 10, line 214: Please add the unit for the reported delta sigma value. Caption Fig. 5 should be directly below the figure, not on the next page. Same for Fig. 1A.

Fig. 2A: Nice figure but please add information why "2 plumes" are visible in certain snapshots (is this related to different wind directions in different altitude?).

References:

Nassar, R., Hill, T. G., McLinden, C. A., Wunch, D., Jones, D., and Crisp, D.: Quantifying CO2 emissions from individual power plants from space, Geophys. Res. Lett., 44, 10045–10053, https://doi.org/10.1002/2017GL074702, 2017.

Reuter, M., Buchwitz, M., Schneising, O., Krautwurst, S., O'Dell, C. W., Richter, A., Bovensmann, H., and Burrows, J. P.: Towards monitoring localized CO2 emissions from space: co-located regional CO2 and NO2 enhancements observed by the OCO-2 and S5P satellites, Atmos. Chem. Phys., https://www.atmos-chem-phys.net/19/9371/2019/, 19, 9371-9383, 2019.

---

## Referee Comment (RC2) · Anonymous Referee #2 · 10 Dec 2020

December 9, 2020

I have reviewed the following paper which is now available for discussion on Atmospheric Measurement Techniques as amt-2020-390:

"Determination of the Emission Rates of CO2 Point Sources with Airborne Lidar," by Sebastian Wolff, Gerhard Ehret, Christoph Kiemle, Axel Amediek, Mathieu Quatrevalet1, Martin Wirth, and Andreas Fix

General comments: The paper describes the use of measurements from the airborne CHARM-F IPDA lidar to estimate the CO2 emission rates from a large European fossil fuel power plant. The measurements were made during the 2018 airborne CoMET campaign. Multiple overflights of the emission plume from the plant were performed,

and the increase in the DAOD of the lidar's XCO2 measurements were determined during the crossings over the plume. From these measurements the authors used the cross-sectional flux method to estimate the power plants rate of CO2 emission. The paper gives overviews of the CHARM-F lidar, the cross-sectional flux method and computes the emission rates from four plume overpasses made on one day. It also describes the impact of turbulent boundary layer mixing on the plume caused by daytime solar heating of the Earth's surface.

To investigate the impact of turbulence further, the authors performed a time-resolved simulation of a modelled plume, which showed the plume's 3-D structure as a function of time of day. These results are quite interesting and clearly show the impact of daytime turbulence on the emission plume structure. The authors discuss the airborne measurement results in the context of the simulations, and the implication of the simulation results on estimates of emission rates for those made using IPDA lidar and those using passive spectrometers.

This paper addresses an important topic, given the importance of remotely sensing CO2 and CH4 emissions to monitor drivers of climate change. Both the airborne measurement results and simulation results are quite interesting. However, there are several issues and questions in the present version that are either unclear or need to be addressed in a revised version. These are briefly discussed below.

Specific Comments: 1. The simulations show that daytime turbulence randomly changes the 3-D velocity field of the plume on short spatial and time scales. Is the cross-sectional method used to compute fluxes still viable under these conditions? The authors need to discuss this point, and if they feel it is, please address how the value and direction of the velocity vector is obtained for overpasses during turbulent conditions. 2. It seems the precision of the lidar estimate of A depends on how many lidar measurements occur during the plume crossing. Since the laser pulse rate is fixed, it seems crossing the plume at an oblique angle (rather than at right angles) should allow more lidar measurements to be used in computing A, hence should increase the precision of A's estimated value. Please discuss. 3. Several places in the paper state that the differential absorption cross section of $CO_2$ for CHARM-F is constant with altitude. For these measurements CHARM-F used the online wavelength locked to the peak of the $CO_2$ line. Due to the decreasing atmospheric pressure with altitude, the absorption cross section of the $CO_2$ line and vertical weighting functions of airborne IPDA lidar which uses this approach is not uniform with altitude. It instead peaks at the aircraft altitude. (For example, see Figure 6 in Bell et al. (2020) "Evaluation of OCO-2 XCO2 variability at local and synoptic scales using lidar and in situ observations from the ACT-America campaigns." Journal of Geophysical Research: Atmospheres, https://doi.org/10.1029/2019JD03140.) Please address this point in the revision. 4. The simulations clearly show the transition of smooth flow of the plume during and nighttime and at low sun angles, to a more chaotic/random dispersed structure caused by turbulence at higher sun angles. This dispersal of the plume's structure seems an important limitation to using passive remote sensing to estimate power plant emissions. Can the authors expand on this point in the revised version? 5. Please add the local sun angles to the figures of the simulated plumes. 6. Equation 1 uses a mixture of laser power and laser energy, and obviously the optical power changes during the pulse. Isn't the DAOD computed from measurements of the on- and off-line laser energies (not power)? 7. In general it seems that the emission plume adds $CO_2$ to an air mass that already has some variability in $CO_2$. Hence the background also has variability in XCO2. As shown in equation 5, the enhancement from the plume is computed from the total DAOD minus that of the background (non-plume region). Hence variability in the background DAOD will cause variability in computed enhancement. More discussion of about the variability in the background is needed for the region measured near the plumes and on its impact on the computed enhancement from the plume. 8. In Figure 3, please inform the reader how many lidar measurements are used in the running means values, and the approximate standard deviations of DAOD for the lidar measurements with and without averaging. 9. In line 238, it is stated the primary error in the flux estimate is from uncertainty in wind speed. This is

an important point and should be emphasized in the paper's abstract and conclusion. 10. Line 238 gives a reported value for power plant's $CO_2$ emission mass flow rate. Please clarify the source of the estimate. Is this some sort of average or is it based on the fossil power plant's operating conditions at the time of the overflights? 11. The caption of Figure 6 states DAOD enhancements < 0.008 cannot be distinguished. It is unclear if the authors mean in the simulation, in the color plot, or in the CHARM-F lidar measurements – please clarify in the text. 12. The LH plot in Figure 6 shows a stable linear plume extending to the edge of the plot for nighttime conditions. It would benefit the readers to know from the simulations typically how far these linear flow conditions extend in distance. 13. In the paragraph of line 285, please explain in the simulation how the plume's velocity values and directions were estimated for the turbulent conditions. 14. Line 330 – Please be more quantitative than "mid-day" please clarify the sun angle limits or time of day limits for avoiding turbulent mixing. 15. In the discussion section, please address how more accurate estimates of the wind vector in the plume could be attained in the future. For example, could co-aligned wind and IPDA lidar be used for this? 16. Several aspects of the last paragraph in the Conclusion section don't seem to apply to the main findings of the paper. Please reexamine and edit for relevance. 17. Figure A2 – please include the sun angles for the plumes shown in daytime hours.

Minor points/technical corrections: 1. The manuscript will benefit from a check of consistency of tenses. The airborne campaign was performed in the past, while the analysis may described in the present tense. 2. In the title, would the word "estimation" perhaps be a more appropriate word than "determination"? 3. Line 17 – Consider changing "we suppose." Did not the simulations show the variability was clearly driven by turbulent mixing ? 4. Line 30: consider perhaps "assessment" instead of "stocktake" 5. Line 43 – this sentence is many lines long. Please break it into logical pieces. 6. Page 2 and elsewhere. There are many adjectives used in the lidar description and elsewhere (e.g. on page 2: most sophisticated, small divergence, dense sequence, sophisticated, high-power etc,) whose meanings are subjective. Please either delete

or replace these type of adjectives with quantitative descriptors. 7. Line 60. The term "adaptive averaging "does not appear to be not defined or described 8. Line 199 – the distances listed in the text are slightly different than in Table 1 9. Table 1- 2nd column – please check, are the crossing dimensions km or in m? 10. Table 1 – is the mean q listed for all crossings? – please clarify 11. Figure 3 would be easier to interpret if the 3 boxes were labelled. 12. Line 365 – please delete the word "high" and replace with the approximate accuracy attained.
* * *

---

## Author Comment (AC1) · 5 Feb 2021

Dear Anonymous Referee #1,

Thank you very much for this comprehensive review. We appreciate the level of detail and your effort very much. All the comments are useful and help improving this work. We answered all questions and implemented your suggestions. Your comment is repeated underneath in **bold font**, answers are written in *italics*, changes regarding the manuscript are written in *blue italics*. In some cases, you referred to specific pages, lines or equations. In the revised version these references might have changed. If applicable we have inserted the new reference in red brackets **[]** with respect to the revised version in change mode.
Please note that the "Data availability" has been moved after the Appendix A.
New references were added in the revised version, as well as in our response here. You can find them at the end of the document.

1. **Abstract, line 8, sentence "Due to the large number of point sources and their global spatial distribution, a mobile measurement approach with fast spatial coverage is needed": Unclear, why a "mobile measurement approach with fast spatial coverage is needed"? Why mobile? Why spatially fast? Regular on-site monitoring could be even better. Please consider to revise this sentence.**
   *We agree, at this point our phrasing is misleading. What we are aiming at is a satellite-based observation system. Referring to the high number of point sources and their global distribution, it is evident that on-site monitoring would entail the use of a high number of measurement instruments, and/or an enormous work effort. Being of course also challenging, satellite missions are convenient for remote monitoring on a global scale. Which is why we rephrased as follows:* *"Due to the large number of point sources and their global spatial distribution, the implementation of a satellite-based observation system is convenient. Airborne active remote sensing measurements demonstrate that the deployment of lidar is promising in this respect."*

2. **Abstract, line 15 [now line 16], sentence "emission rates can be derived in terms of the cross–sectional flux method". I recommend to replace "in terms of" by "via" or equivalent.**
   *The suggested replacement has been incorporated:* *"From the resulting enhancement in the $CO_2$ mixings ratios, emission rates can be derived via the cross–sectional flux method."*

3. **Page 2, line 37 [now line 39], sentence "However, at the moment no operating satellite mission is able to reliably quantify emissions from large power plants." Unclear what exactly is meant here taking into account publications such as Nassar et al., 2017, and Reuter et al., 2019, which need to be cited (details see below in "References").**
   *We agree and have cited the references in the revised version. We have rephrased as follows:* *"Under particularly favorable conditions, it is already possible to detect $CO_2$ emissions of power plants from space, as is done with data from NASA's OCO-2 mission (Nassar et al., 2017; Reuter et al., 2019). However, at the moment no operating satellite mission is able to quantify emissions from large power plants, on a regular basis."*

4. **Page 2, line 60 [now line 66], sentence "Albedo variations basically influence the measurement precision (statistical uncertainty), not the bias.": This is a bold statement. Is it really true that the impact of albedo variation on the bias is zero or are corresponding biases only expected to be very small?**
   *Admittedly, as the retrieval involves a non-linear computational operation, the influence on the bias is still negligible, but, strictly speaking, not exactly zero. We have changed the sentence to* *"Albedo variations basically affect the measurement precision (statistical uncertainty), whereas the influence on the bias is negligible (Amediek et al., 2009)."*

5. **Page 3, line 67 [now line 75], sentence "This principle has been applied for : : :". The cross-sectional flux method has also been applied to satellite data, see Reuter et al., 2019. Please add this missing information.**
*The suggested reference has been added*

6. **Page 2 [now page 5], eq (2): Reader may wonder why delta sigma, the absorption cross section difference, does not depend on altitude. Later this aspect is addressed but I recommend to add some details on this aspect when presenting Eq 2.**
*To simultaneously address a comment from Reviewer #2, we have merged Eq. (2) with the former Eq. (5). Eq. (2) itself now reads: "$DAOD = DAOD_b + \Delta DAOD = DAOD_b + \frac{1}{M}\int_0^{fl} \Delta\sigma(z) \cdot \Delta c(z)\, dz \approx DAOD_b + \frac{\overline{\Delta\sigma}}{M}\int_0^{fl} \Delta c(z)\, dz$"*
*Subsequent to Eq. 2 [now line 115], we have rephrased as follows: "$\Delta\sigma(z)$ is the difference between the absorption cross section of the two laser pulses given in square meter (cf. Fig. 1). It is referred to as the differential–absorption cross section. Generally, $\Delta\sigma(z)$ is not constant over the plume's vertical extension, due to the decreasing pressure with altitude. However, the decreases in pressure associated with typical vertical plume extensions are small. As an approximation we use the mean value over the vertical extent of the plume $\overline{\Delta\sigma}$. This aspect is discussed in more detail in Sect. 3."*

7. **Page 6 , line 129 [now page 7 line 143]: Reference to Strandgren et al., 2020: Unclear to which statement or fact the reference refers to. Please also add space between "2020)driven".**
*The reference was inserted by accident. It has been removed.*

8. **Page 7, line 154 [now page 9 line 173], sentence "For the definition of the plume's limits a running mean with such a width must be used that the plume enhancement is blurred." Not mandatory but please consider to improve this sentence.**
*In this passage we have changed a little more for better readability. The two previous sentences have been moved up to where the running mean width of 0.2 km is first mentioned [line 166 and preceding]: "For this we examine a 0.2 km running mean of the DAOD dataset (Fig. 3a). The choice of 0.2 km is made because it corresponds to the diameter of the pixels of the simulation (see Sect. 4)."*
*In line 173 and subsequent we now write: "Applying a running mean broadens and flattens the plume. For larger running mean widths, as for example 4 km, the flattening is so severe that the plume is only distinguishable from the background as a raised plateau (see Fig. 3b)."*

9. **Page 7, line 156 [now page 10 line 177], sentence: "Further on the data within the limits : : :". Please improve this sentence.**
*The sentence has been removed.*

10. **Page 9, Figure 4, [now page 11] right (photo): I wonder if the authors have the appropriate right to use the photo. Please confirm.**
*The photograph was taken by Co-Author Andreas Fix.*

11. **Page 10, line 213 [now page 12 line 236]: What is "surface scattering elevation"? Just the surface elevation of the ground scene?**
*Yes, in this context it is simply the height of the ground. In general, it refers to the elevation of the backscattering surface, which could for example also be clouds. For better comprehensibility we have changed the sentence to: Consequently the differential-absorption cross section at the height of the ground (70 m a.s.l.), corresponds to $\Delta\sigma(z = 0\ m) = 7.10652\cdot10\text{-}27\ m^2$.*

**12. Page 10, line 214 [now page 12 line 237]: Please add the unit for the reported delta sigma value.**
*The unit has been added in page 12 line 237, as well as in Eq.(8) page 12 line 235.*

**13. Caption Fig. 5 should be directly below the figure, not on the next page. Same for Fig. 1A.**
*Fig. 5 and Fig. A1 were merged with their respective caption on one side. To do so, Fig. 5 and Fig. 6 were repositioned within the text, and Fig. A1 was reduced in size.*

**14. Fig. 2A: Nice figure but please add information why "2 plumes" are visible in certain snapshots (is this related to different wind directions in different altitude?).**
*Precisely. We've appended an explanation to the caption: "Some snapshots show disjointed exhaust plumes. This is due to vertical wind shear and the resulting different vertical advection directions."*

**References:**

Amediek, A., Fix, A., Ehret, G., Caron, J., and Durand, Y.: Airborne lidar reflectance measurements at 1.57 mu m in support of the A-SCOPE mission for atmospheric CO2, Atmos Meas Tech, 2, 755-772, https://doi.org/10.5194/amt-2-755-2009, 2009.

Nassar, R., Hill, T. G., McLinden, C. A., Wunch, D., Jones, D. B. A., and Crisp, D.: Quantifying CO2 Emissions From Individual Power Plants From Space, 44, 10,045-010,053, https://doi.org/10.1002/2017GL074702, 2017.

Reuter, M., Buchwitz, M., Schneising, O., Krautwurst, S., O'Dell, C. W., Richter, A., Bovensmann, H., and Burrows, J. P.: Towards monitoring localized CO2 emissions from space: co-located regional CO2 and NO2 enhancements observed by the OCO-2 and S5P satellites, Atmos. Chem. Phys., 19, 9371-9383, https://doi.org/10.5194/acp-19-9371-2019, 2019.

---

## Author Comment (AC2) · 5 Feb 2021

Dear Anonymous Referee #2,

Thank you very much for this comprehensive review. We appreciate the level of detail and your effort very much. All the comments are useful and help improving this work. We answered all questions and implemented your suggestions. Your comment is repeated underneath in **bold font**, answers are written in *italics*, changes regarding the manuscript are written in *blue italics*. In some cases, you referred to specific pages, lines or equations. In the revised version these references might have changed. If applicable we have inserted the new reference in red brackets **[]** with respect to the revised version in change mode.

Please note that Sect. 7 "Conclusion and outlook" has been split up in the revised version into Sect. 7 "Conclusion" and Sect. 8 "Outlook". Moreover, "Data availability" has been moved after Appendix A. New references were added in the revised version, as well as in our response here. You can find them at the end of the document.

Specific Comments:

1. **The simulations show that daytime turbulence randomly changes the 3-D velocity field of the plume on short spatial and time scales. Is the cross-sectional method used to compute fluxes still viable under these conditions? The authors need to discuss this point, and if they feel it is, please address how the value and direction of the velocity vector is obtained for overpasses during turbulent conditions.**

   *An assumption that must be fulfilled for the cross-sectional flux method is, that the tracer must have an advection component perpendicular to the overflight, i.e. u in Eq. (3). In other words, there must be a flux through the lidar curtain. In particular, Eq. (4) shows that the error in the emission rate calculation becomes very large when this wind component has small values. Varon et al. (2018) impose 2 m/s as the minimum wind speed required for the applicability of the cross-sectional flux method. This minimum value is also used by Sharan et al. (1996), explaining that above this threshold, advection dominates over diffusion. At the end of Sect. 2.1 we have therefore appended:* "Varon et al. (2018) have identified 2 m/s as the minimum threshold of wind speed for the applicability of the cross-sectional flux method. This minimum value is also referred to by Sharan et al. (1996), arguing that above this threshold advection dominates over diffusion."

   *The mean wind speed observed for the measurement is well above this threshold. Below table 1 we have inserted a remark to this regard:* "The mean wind speed is well above the threshold of 2 m/s, introduced at the end of Sect. 2.1."

   *This is also the case for the simulation. We have added the sentence* "The wind speed u surpasses the threshold value of 2 m/s at all times." *to the caption of Fig. 8.*

   *A description of the retrieval of the wind vector can be found in the last passage of Sect. 4. The described procedure is used under all conditions.*

2. **It seems the precision of the lidar estimate of A depends on how many lidar measurements occur during the plume crossing. Since the laser pulse rate is fixed, it seems crossing the plume at an oblique angle (rather than at right angles) should allow more lidar measurements to be used in computing A, hence should increase the precision of A's estimated value. Please discuss.**

   *It is true that a higher number of measured data points within the plume would increase the precision of A's estimated value. However, under consideration of the error budget estimated using Eq. (4) it becomes evident, that this would increase the overall error in the emission rate estimation. To illustrate this, we plotted the error δq/q over the crossing angle φ for the four measurement overflights, using the values given in Table 1:*

[Figure]

*The dashed red vertical line marks the angle of HALO's flight track to the mean wind direction.*

3. **Several places in the paper state that the differential absorption cross section of CO2 for CHARM-F is constant with altitude. For these measurements CHARM-F used the online wavelength locked to the peak of the CO2 line. Due to the decreasing atmospheric pressure with altitude, the absorption cross section of the CO2 line and vertical weighting functions of airborne IPDA lidar which uses this approach is not uniform with altitude. It instead peaks at the aircraft altitude. (For example, see Figure 6 in Bell et al. (2020) "Evaluation of OCO-2 XCO2 variability at local and synoptic scales using lidar and in situ observations from the ACT-America campaigns." Journal of Geophysical Research: Atmospheres, https://doi.org/10.1029/2019JD03140.) Please address this point in the revision.**

*Our measurements are based on two laser pulses, which are very well constrained in wavelength. For our purpose, we are only interested in their differences in optical depth DAOD, or more specifically, only in the DAOD enhancement induced by the plume. To clarify this, we have modified Eq. (2), thereby harmonizing it with Appendix A in Amediek et al. (2017). Note that Eq. (5) has therefore been removed.*

*It is true that the differential-absorption cross section $\Delta\sigma$ is not constant vertically, but, in the integral of Eq. (2), we are focusing on the vertical range in which the plume provides enhanced CO2 density, compared to the background concentration. The vertical extension of the plume at crossing points will be estimated by means of various atmospheric stability conditions, that can be found in the literature, as discussed in more detail in Sect. 3. It turned out that the change in $\Delta\sigma$ over the vertical plume spread is small compared to the other error sources. So, it is regarded as legitimate to use the mean $\Delta\sigma$, which simplifies the calculation immensely (see Appendix A in Amediek et al. (2017)). We have replaced the term "constant" by "mean" and consistently marked $\overline{\Delta\sigma}$ with an overscore in the entire text, to make the distinction recognizable. The fact that the mean differential-absorption cross section is only an approximation, is widely discussed in our error calculation in Sect. 3 at Eq. (8) and surrounding passages.*

*To convert DAOD to XCO2, the integrated vertical weighting function would be required. However, this is not necessary for the calculation of the emission rate. Therefore, unlike in Amediek et al. (2017), we do neither need to consider XCO2 nor the vertical weighting function in this work.*

4. **The simulations clearly show the transition of smooth flow of the plume during and nighttime and at low sun angles, to a more chaotic/random dispersed structure caused by turbulence at higher sun angles. This dispersal of the plume's structure seems an important**

**limitation to using passive remote sensing to estimate power plant emissions. Can the authors expand on this point in the revised version?**

*We fully agree, which is why we've highlighted this aspect in the conclusion. The independence from solar irradiation is an advantage of active remote sensing to this respect. However, we are no experts for passive remote sensing and also our modeling efforts have focused on IPDA-lidar. CoMet saw the deployment of EM27 Fourier transform spectrometers, as well as the airborne passive instrument MAMAP. Synergistic studies will follow in the future; for the time being, we refer to Luther et al. (2019) and Krautwurst et al. (2021).*

5. **Please add the local sun angles to the figures of the simulated plumes.**

   *The local solar altitudes α have been added. We've also inserted the course of the solar altitude to Figure 8a and comment as of* line 323*:* "In Fig. 8a it can be seen how the diurnal course of solar altitude α influences the retrieved emission rates q. The random occurrence of inhomogeneities in the plume propagation, caused by local turbulence, leads to large variations in the results of successive crossings. Turbulence lags behind solar altitude because the surface needs time to heat up. It is also apparent that the emission rate deviations vary from day to day, both in intensity, as well as in dwell time.
   The implications for the measurement results can be reduced by averaging over a multitude of retrieved emission rates. Next, we investigate how often the exhaust plume must be surveyed, in order to achieve a mean emission rate with satisfactory accuracy."

6. **Equation 1 uses a mixture of laser power and laser energy, and obviously the optical power changes during the pulse. Isn't the DAOD computed from measurements of the on- and off-line laser energies (not power)?**

   *In the lidar community, the term "power" is used quite laxly, although this is not accurate. In fact, it is the radiation fluxes entering the lidar telescope. We have added the reference Ehret et al. (2008), where the lidar equation for the "hard target" case is described. In this equation, the received radiation fluxes $P_{on/off}$ are proportional to the emitted pulse energy $E_{on/off}$ divided by an effective pulse length in the temporal domain of the lidar returns. This effective pulse length is regarded to be equal for both on- and off-line wavelengths and therefore cancels out by taking the ratio in Eq. (1). But the latter is not the case for the pulse energies, which are different for both wavelengths and therefore need to be considered in Eq. (1). More details to the shape of the lidar returns can be found in Ehret et al. (2008). For the measurement we digitally oversample the lidar returns in the temporal domain sufficiently well, which allows us to calculate both the corresponding radiation flux and the range to the targets on a shot-by-shot basis very accurately.*

7. **In general it seems that the emission plume adds CO2 to an air mass that already has some variability in CO2. Hence the background also has variability in XCO2. As shown in equation 5** [now Eq. (2)]**, the enhancement from the plume is computed from the total DAOD minus that of the background (non-plume region). Hence variability in the background DAOD will cause variability in computed enhancement. More discussion of about the variability in the background is needed for the region measured near the plumes and on its impact on the computed enhancement from the plume.**

   *It is true, that the plume adds CO2 to a variable background, thereby inducing a strong local gradient in the DAOD. This plume enhancement is much more significant than variability in the background, which is why the plume can easily be located. With regard to the background, we've introduced a procedure in Sect. 2.2 that allows the background term to feature gradients on the scale of few hundreds of meters. These small-scale gradients in the background term are not nearly as pronounced as the plume gradient, or even the noise for that matter. Figure 3d shows that the background term is not constant. Here is Figure 3 again, zoomed in for better visibility:*

[Figure]

Evidently, our procedure does not attribute spatially smaller gradients to the background term, but incorporates them in the enhancement term. Thereby they are not distinguishable from noise and basically add an uncertainty in the estimation of the integrated enhancement A. However, we consider this circumstance to be minor compared to the actual noise.

In the caption of Figure 3 we've rephrased as follows: *"In (d) again a 4 km running mean over the bypassed dataset is shown in brown. This data, which has slight variability, is used as background term DAOD_b."*

Furthermore, we've added some explanatory sentences to the passage below the figure *[line 179]*: *"At last, we execute another 4 km running mean over the raw dataset, with bypassed plumes, resulting in the background term DAOD_b, shown in brown in Fig. 3d. This procedure allows for a variability in the background term on a scale of a few hundred meters. Smaller scale gradients cannot be attributed to the background and are incorporated in the enhancement term ΔDAOD, thereby not being distinguishable from noise."*

8. **In Figure 3, please inform the reader how many lidar measurements are used in the running means values, and the approximate standard deviations of DAOD for the lidar measurements with and without averaging.**

   *In the caption we now write:* *"Figure 3: Plume crossing at a point source distance of 1.53 km. In (a) the gray curve shows the raw data, with a standard deviation of 5.2 %, while the black curve shows a 0.2 km (64 data points) running mean (RM), with a standard deviation reduced to 0.9 %. In (b) the green curve is a 4 km (1293 data points) RM. Green vertical dashed […]"*

9. **In line 238 [now 263], it is stated the primary error in the flux estimate is from uncertainty in wind speed. This is an important point and should be emphasized in the paper's abstract and conclusion.**

   *We agree and added to the abstract in line 17:* *"On average, our results roughly correspond to reported annual emission rates, with wind speed uncertainties being the major source of*

*error. We observe significant variations between individual overflights, ranging up to a factor of 2."*

*In Sect. 7 Conclusion [line 411] we write: "On the one hand, this is because the uncertainties in the wind speed are most pronounced at these times, being the major source of error in a single measurement"*

10. **Line 238 [now 264] gives a reported value for power plant's Co2 emission mass flow rate. Please clarify the source of the estimate. Is this some sort of average or is it based on the fossil power plant's operating conditions at the time of the overflights?**

    *It is the mean value for the year 2017, reported to the European Environment Agency. This mean value is introduced in the beginning of Sect. 3. For clarification we've inserted the reference. Line 254 now reads: "The reported value of 760 kg(CO2)/s (24.0 Tg(CO2)/yr) (E-PRTR, 2020) lies within the error range of [...]"*

11. **The caption of Figure 6 states DAOD enhancements < 0.008 cannot be distinguished. It is unclear if the authors mean in the simulation, in the color plot, or in the CHARM-F lidar measurements – please clarify in the text.**

    *In the caption now reads: "In a corresponding measurement, DAOD enhancement values beneath 0.008 would not be distinguishable from noise and are therefore displayed blue."*

12. **The LH plot in Figure 6 shows a stable linear plume extending to the edge of the plot for nighttime conditions. It would benefit the readers to know from the simulations typically how far these linear flow conditions extend in distance.**

    *As a matter of fact, the linear plume is even extending the edge of the inner domain. That is why we can only report that the linear flow exceeds a distance of 17.5 km, i.e. half the extent of the domain.*

13. **In the paragraph of line 285 [now 310], please explain in the simulation how the plume's velocity values and directions were estimated for the turbulent conditions.**

    *A description of the retrieval of the wind vector can be found right above, in the last passage of Sect. 4. The described procedure is used under all conditions.*

14. **Line 330 [now 357] – Please be more quantitative than "mid-day" please clarify the sun angle limits or time of day limits for avoiding turbulent mixing.**

    *Unfortunately, this is not possible for us. At comment 5. we have already mentioned how we observe different intensities of turbulence day by day. The specific limits in solar altitude/time of day, as well as the synoptic conditions to which they are subject, have to be evaluated by a simulation of a representative number of days in the seasonal course. As we have only performed simulations for our measurement day and the day before, this is not in the scope of this work. Nevertheless, for the two simulated days, we observe, as is to be expected, that the most pronounced turbulence occurs in the afternoon.*

    *We've replaced the wording "midday turbulence" by the more general term "situations of high turbulence". We feel that a more detailed engagement on this issue is better suited in the discussion Sect. 6. There, we elaborate on this matter: "At this point, we cannot derive any limits for solar altitude or local times that should be avoided, as the simulation reveals that the turbulence intensity varies from day to day (see Fig. 8 and Fig. 9). Generally, we find that the most significant turbulence occurs in the afternoon. For future campaign planning, we recommend to also perform measurements at night or in the morning, which is possible with lidar."*

15. **In the discussion section, please address how more accurate estimates of the wind vector in the plume could be attained in the future. For example, could co-aligned wind and IPDA lidar be used for this?**

*An airborne deployment of both wind and IPDA lidar appears to be appealing. Wind lidars perform conical scans, which need up to a minute to be completed. Due to the conical scan, the lidar is pin-pointing transversal to the track of flight. Thus, the retrieved wind speed might have an additional representativeness error, depending on the inhomogeneity of the atmosphere. So, just like in the paper, the measurement result can only yield an average wind vector of the crossings vicinity and not the instantaneous velocity vector of the plume's flux through the lidar curtain. According to our DLR colleagues Benjamin Witschas and Stephan Rahm our in-house airborne wind lidar, provides data with a horizontal resolution of about 9 km and a vertical resolution of 100 m (Witschas et al., 2017). For our application it is possible to reduce the number of line-of-sights. Additionally, a new, faster scanner will be developed. All together it is realistic to achieve a horizontal resolution of less than 4 km in the near future.*

*Whether co-aligned wind lidar measurements are preferable to simulation data for our purpose will be subject of upcoming investigations. For instance, the MAGIC campaign will take place in 2021, during which CHARM-F will be deployed together with one of our partners' airborne wind lidar.*

*Concerning the CoMet campaign, three ground-based doppler wind lidar were installed in the Upper Silesian Coal Basin (USCB) and continuously performed velocity-azimuth display scans. In future studies, dedicated to the USCB, it is planned to nudge the model towards the wind lidar data, as has been done in Kostinek et al. (2020). We've appended to the first passage of the discussion: "Future studies will examine CHARM-F measurements in the Upper Silesian Coal Basin to determine $CH_4$ emissions from coal mines. In this area, ground-based Doppler wind lidars have been installed. It is expected that nudging the simulation towards the wind soundings will result in an improvement of the wind vector estimation, ultimately reducing the overall error in the flux determination."*

*Kostinek et al. (2020) was published for discussion only after our submission. We have included a citation in line 52 of the Introduction.*

16. **Several aspects of the last paragraph in the Conclusion section don't seem to apply to the main findings of the paper. Please reexamine and edit for relevance.**
*Originally, we intended to put the conclusion and outlook in one Section. We now inserted a last Sect. 8 for the outlook, containing the last paragraph. Note that to do so, we've changed the last sentence [now line 93] in the introduction to: "A discussion is given in Sect. 6, followed by the conclusion and outlook can be found in Sect. 7 and the outlook in Sect. 8."*

17. **Figure A2 – please include the sun angles for the plumes shown in daytime hours.**
*The solar altitudes have been included and mentioned in the caption*

Minor points/technical corrections:
1. **The manuscript will benefit from a check of consistency of tenses. The airborne campaign was performed in the past, while the analysis may described in the present tense.**
*We have carried out the check. With a few exceptions, the tenses were consistent, however, we inserted several commas and corrected the phrasing in multiple passages.*

2. **In the title, would the word "estimation" perhaps be a more appropriate word than "determination"?**
*Indeed, initially we report on an emission rate estimation, with emphasis on a full error assessment. However, subsequently a considerable part of the paper is devoted to the LES simulation. This simulation is by no means used for the estimation of the power plant's emission rate, but rather seeks to provide insights for the determination of point source emission rates using cross-sectional flux method in general. Therefore, we feel that the term "determination" is better suited to encompass the content of the entire work.*

3. **Line 17 [now line 18] – Consider changing "we suppose." Did not the simulations show the variability was clearly driven by turbulent mixing?**
   *At that point, our intention was to hypothesize based on the findings of the measurements. This hypothesis is then confirmed by the simulation. But indeed, the expression "suppose" is inapplicable. We have changed the wording, also for the subsequent sentence t: "We hypothesize that these variations are mostly driven by turbulence. This is confirmed by a high–resolution large eddy simulation that enables us to give a qualitative assessment of the influence of plume inhomogeneity on the cross–sectional flux method."*

4. **Line 30 [now line 32]: consider perhaps "assessment" instead of "stocktake"**
   *We would like to advocate staying with the term "global stocktake", as this is the key word that was originally used in Article 14 of the Paris Climate Agreement. (see https://unfccc.int/sites/default/files/english_paris_agreement.pdf)*

5. **Line 43 [now line 46] – this sentence is many lines long. Please break it into logical pieces.**
   *The sentence has been split into: "The objective of CoMet is to investigate the fluxes of the major human-influenced GHG on local, regional, and sub–continental scales. These fluxes are to be determined more precisely than previously possible. Furthermore, supporting activities for GHG stocktaking are provided."*

6. **Page 2 and elsewhere. There are many adjectives used in the lidar description and elsewhere (e.g. on page 2: most sophisticated, small divergence, dense sequence, sophisticated, high-power etc,) whose meanings are subjective. Please either delete or replace these type of adjectives with quantitative descriptors.**
   *The adjectives "sophisticated" and "high-power" have been removed. In line 98 we now write: "At its core, CHARM-F consists of a pulsed, tunable laser source and a detector."*
   *In the introduction we have changed the wording as follows: "As a result of the pulse repetition frequency (50 Hz, double pulse) and divergence (~1.5 mrad) the pattern on the ground is a sequence of overlapping footprints."*

7. **Line 60 [now 68]. The term "adaptive averaging "does not appear to be not defined or described**
   *High albedo yields a high signal-to-noise-ratio (SNR), while a low albedo yields a poor SNR. By applying a running mean, we can generally improve the measurement precision. However, when we measure between clouds the averaging window is limited by the size of the cloud gap. Performing a running mean that requires adapting the smooth width is what we refer to as adaptive averaging. Be that as it may, for the dataset in hand the SNR does not vary much and clouds do also not play a role. Therefore, this piece of information is irrelevant for this work, which is why we have removed the respective sentence.*

8. **Line 199 [now line 222 and 223] – the distances listed in the text are slightly different than in Table 1**
   *At this point we are dealing with the calculation of the differential-absorption cross section $\Delta\sigma$. Three of the four overflights have taken place close to the point source and one further away. For the calculation of $\Delta\sigma$ the rough distinction between these distances is sufficient, as the change in $\Delta\sigma$ is only minor. In line 222 and 223 we have changed "=" to "≈".*

9. **Table 1- 2nd column – please check, are the crossing dimensions km or in m?**
   *The crossing dimension is km. "path" corresponds to the flown distance of the aircraft (i.e. the cumulative sum of the distances between successive data points). In Figure A1 the term "flight track" is used. For conformity and comprehensibility, the term "path" in the $2^{nd}$ column of table 1 has been changed to "flight track".*

10. **Table 1 – is the mean q listed for all crossings? – please clarify**
   *The mean q is the average of the of the four crossings. For comprehensibility, the columns have been rearranged and a curly bracket has been inserted.*

11. **Figure 3 would be easier to interpret if the 3 boxes were labelled.**
   *We assume Figure 5 showing the nested domains of the simulation was supposed to be addressed with this comment. We added labels to the domains in the Figure.*

12. **Line 365 [now 395] – please delete the word "high" and replace with the approximate accuracy attained.**
   *The sentence has been changed to: "Under such conditions even a single instantaneous cross-sectional flux measurement yields an accuracy of up to ~95 %."*

**References:**

Amediek, A., Ehret, G., Fix, A., Wirth, M., Budenbender, C., Quatrevalet, M., Kiemle, C., and Gerbig, C.: CHARM-F-a new airborne integrated-path differential-absorption lidar for carbon dioxide and methane observations: measurement performance and quantification of strong point source emissions, Appl Optics, 56, 5182-5197, https://doi.org/10.1364/AO.56.005182 2017.

Ehret, G., Kiemle, C., Wirth, M., Amediek, A., Fix, A., and Houweling, S.: Space-borne remote sensing of CO2, CH4, and N2O by integrated path differential absorption lidar: a sensitivity analysis, Applied Physics B, 90, 593-608, https://doi.org/10.1007/s00340-007-2892-3, 2008.

Kostinek, J., Roiger, A., Eckl, M., Fiehn, A., Luther, A., Wildmann, N., Klausner, T., Fix, A., Knote, C., Stohl, A., and Butz, A.: Estimating Upper Silesian coal mine methane emissions from airborne in situ observations and dispersion modeling, Atmos. Chem. Phys. Discuss., 2020, 1-24, https://doi.org/10.5194/acp-2020-962, 2020.

Krautwurst, S., Gerilowski, K., Borchardt, J., Wildmann, N., Galkowski, M., Swolkien, J., Marshall, J., Fiehn, A., Roiger, A., Ruhtz, T., Gerbig, C., Necki, J., Burrows, J. P., Fix, A., and Bovensmann, H.: Quantification of CH4 coal mining emissions in Upper Silesia by passive airborne remote sensing observations with the MAMAP instrument during CoMet, Atmos. Chem. Phys. Discuss., 2021, 1-39, https://doi.org/10.5194/acp-2020-1014, 2021.

Luther, A., Kleinschek, R., Scheidweiler, L., Defratyka, S., Stanisavljevic, M., Forstmaier, A., Dandocsi, A., Wolff, S., Dubravica, D., Wildmann, N., Kostinek, J., Jöckel, P., Nickl, A. L., Klausner, T., Hase, F., Frey, M., Chen, J., Dietrich, F., Nęcki, J., Swolkień, J., Fix, A., Roiger, A., and Butz, A.: Quantifying CH4 emissions from hard coal mines using mobile sun-viewing Fourier transform spectrometry, Atmos. Meas. Tech., 12, 5217-5230, https://doi.org/10.5194/amt-12-5217-2019, 2019.

Sharan, M., Yadav, A. K., Singh, M. P., Agarwal, P., and Nigam, S.: A mathematical model for the dispersion of air pollutants in low wind conditions, Atmospheric Environment, 30, 1209-1220, https://doi.org/10.1016/1352-2310(95)00442-4, 1996.

Varon, D. J., Jacob, D. J., McKeever, J., Jervis, D., Durak, B. O. A., Xia, Y., and Huang, Y.: Quantifying methane point sources from fine-scale satellite observations of atmospheric methane plumes, Atmos. Meas. Tech., 11, 5673-5686, https://doi.org/10.5194/amt-11-5673-2018, 2018.

Witschas, B., Rahm, S., Dörnbrack, A., Wagner, J., and Rapp, M.: Airborne Wind Lidar Measurements of Vertical and Horizontal Winds for the Investigation of Orographically Induced Gravity Waves, Journal of Atmospheric and Oceanic Technology, 34, 1371-1386, https://doi.org/10.1175/JTECH-D-17-0021.1, 2017.

---

## Author Response (AR2)

Dear Andreas Richter,

Thank you very much for your suggestions.

• L11: "weighted vertical columns of CO2 mixing ratios" - I'm confused by this formulation. To me, the result of the measurements are vertically averaged CO2 mixing ratios, not columns

We have changed the wording to: "The integrated-path differential-absorption lidar CHARM–F is installed onboard an aircraft, in order to detect weighted column-integrated dry-air mixing ratios of  $CO_2$ ." We use the word "column" to indicate that the vertical integration limits correspond to the scanned volume beneath the aircraft, not the entire atmosphere. For clarification we refer to Eq. (3) in Ehret et al. 2017 (https://doi.org/10.3390/rs9101052) descriptive for the satellite mission MERLIN and therefore methane, but of course equivalent for  $CO_2$ :

$$XCH_{4} \equiv \frac{\int_{0}^{p_{T}} m_{r}(p)WF(p,T)dp}{\int_{0}^{p_{T}} WF(p,T)dp} = \frac{DAOD}{\int_{0}^{p_{T}} WF(p,T)dp} = \frac{\ln\left[\frac{P_{off}(r_{T})E_{on}}{P_{on}(r_{T})E_{off}}\right] - DAOD_{other \ gases}}{2\int_{0}^{p_{T}} WF(p,T)dp}$$

where

$$WF(p,T) = \frac{\sigma_{on}(p,T) - \sigma_{off}(p,T)}{g(p)M_{air}(1 + M_{H_2O}q(p)_{air})}$$

 $m_r$  being the dry-air mixing ratio of CH4 at a given pressure p,  $p_T$  being the pressure at the reflecting surface (target) and  $q_{dry}$  is the water vapor mixing ratio with respect to dry air.

- L95: molecular, absorption => molecular absorption Emended
- L107: "Δσ(z) is the difference between the absorption cross section of the two laser pulses given in square meter" not entirely precise as the cross-section is the one of CO2, not of the laser pulse and the difference is because of the difference in wavelength. What about "Δσ(z) is the difference between the CO2 absorption cross section for the two laser wavelengths given in square meter"? Emended
- L107: Cross-section is not explicitly mentioned in the Figure, just "absorption" At this point, the Figure depicts the principle of an IPDA lidar schematically. A high cross-section corresponds to a high absorption, while a low cross-section corresponds to a low absorption. In the Caption we've changed "absorption wavelength" to "absorption line", as a molecule doesn't have a wavelength. Moreover, we've appended the following sentence to the caption: "The black line in (b) schematically depicts the measurement principle, not the actual spectral absorption line shape of CO2." If there are no objections from your side, we would like to stick to the label "absorption".
- Caption Figure 4: "and horizontal spacing of 250 m and 50 m" not sure how to interpret this It's the distance between the individual towers. We've changed the sentence to: "The towers have a height of 120 m and distances of 250 m and 50 m between each other."
- L228: Errors according to => Errors resulting from Emended
- L230: need not be => need not to be Emended
- L242 and elsewhere: two-tenth: I find this way of separating the error into its components quite unusual, why not say "20% of the uncertainty..."?

The intention was to not go into the quantitative details of the values of the error components, but to outline their rough attribution fraction. In this way, we thought, we'd guide the reader most straight forward, to the importance of the error attribution by the wind. Moreover, we intended to prevent the reader from confusing the error fractions with the actual values of the errors, since we also give these as percentages in several places.

Be that as it may, we consent that this is an unusual way, but feel it is constructive at this point. To make the attribution more visually accessible, we have changed the written-out attributions into fractional numbers. We would like to keep it this way, if you disagree, please let us know.

- Section 4: I'm not sure what the right way of determining the mean wind in the model results really is. For a • realistic comparison to the measurements, the right wind would be that of the model run at lower resolution. I would assume, that strong turbulence will lead to strong fluctuations in wind speed and direction over time and altitude, and in a real world application, this will further increase the uncertainty of the assumed wind vector and thus the emission estimate. My concern is, that by trying to find the best wind vector from the simulation, the uncertainty is actually underestimated.

Measurement uncertainties in a real-world application would certainly add up to that. However, the purpose of our assessment is not to provide an uncertainty estimate. We want to pinpoint to the variations in the retrieved emission rates that arise from inhomogeneities induced by turbulence alone and the potential of avoiding times of strong turbulence. In this consideration, we have tried to keep systematic errors to a minimum. This is the reason why we do not simulate measurement noise, for that matter.

L373: higher number of crossings are required => larger number of crossings is required Emended

All individual changes can be reviewed in the marked-up manuscript version below.

On behalf of all Co-Authors, Sebastian Wolff

[revised manuscript text omitted]